# Fast Scalable and Accurate Discovery of DAGs Using the Best Order Score Search and Grow-Shrink Trees

**Bryan Andrews**
Department of Psychiatry & Behavioral Sciences
University of Minnesota
Minneapolis, MN 55454
andr1017@umn.edu

**Joseph Ramsey**
Department of Philosophy
Carnegie Mellon University
Pittsburgh, PA 15213
jdramsey@andrew.cmu.edu

**Rubén Sánchez-Romero**
Center for Molecular and Behavioral Neuroscience
Rutgers University
Newark, NJ 07102
ruben.saro@rutgers.edu

**Jazmin Camchong**
Department of Psychiatry & Behavioral Sciences
University of Minnesota
Minneapolis, MN 55454
camch002@umn.edu

**Erich Kummerfeld**
Institute for Health Informatics
University of Minnesota
Minneapolis, MN 55454
erichk@umn.edu

## Abstract

Learning graphical conditional independence structures is an important machine learning problem and a cornerstone of causal discovery. However, the accuracy and execution time of learning algorithms generally struggle to scale to problems with hundreds of highly connected variables—for instance, recovering brain networks from fMRI data. We introduce the best order score search (BOSS) and grow-shrink trees (GSTs) for learning directed acyclic graphs (DAGs) in this paradigm. BOSS greedily searches over permutations of variables, using GSTs to construct and score DAGs from permutations. GSTs efficiently cache scores to eliminate redundant calculations. BOSS achieves state-of-the-art performance in accuracy and execution time, comparing favorably to a variety of combinatorial and gradient-based learning algorithms under a broad range of conditions. To demonstrate its practicality, we apply BOSS to two sets of resting-state fMRI data: simulated data with pseudo-empirical noise distributions derived from randomized empirical fMRI cortical signals and clinical data from 3T fMRI scans processed into cortical parcels. BOSS is available for use within the TETRAD project which includes Python and R wrappers.

## 1   Introduction

We present a permutation-based algorithm, Best Order Score Search (BOSS), for learning a Markov equivalence class of directed acyclic graphs (DAGs). A novel method for caching intermediate calculations for permutation-based algorithms, Grow-Shrink Trees (GSTs), is also developed and presented in this paper. Our implementation of BOSS using GSTs scales well in both accuracy and time to higher numbers of variables and graph densities. We demonstrate that in particular, this method scales at least to the complexity of dense cortical parcellations of fMRI data.

37th Conference on Neural Information Processing Systems (NeurIPS 2023).

In real world systems, it is not unusual to have hundreds or thousands of variables, each of which may be causally connected to dozens of other variables. Models of functional brain imaging (fMRI), functional genomics, electronic health records, and financial systems are just a few examples. In order for structure learning methods to have an impact on these real world problems, they need to be not only highly accurate but also highly scalable in both dimensions.

This work follows earlier research on permutation-based structure learning [12, 22, 24, 26]. A previous permutation-based method, Greedy Relaxations of the Sparsest Permutation (GRaSP) [12], demonstrated strong performance on highly connected graphs, but struggled to scale to sufficiently large numbers of variables. By using BOSS and GSTs, we are able to overcome this challenge, while maintaining nearly identical performance to GRaSP. As our simulations show, other popular methods fall short on accuracy, scalability, or both, for such large, highly connected models.

In the remainder of this paper, we provide background for our approach in Section 2, followed by a discussion of GSTs in Section 3. We then introduce BOSS in Section 4 and validate it by comparing it to several alternatives. We show that BOSS using GSTs compares favorably to a number of best-performing combinatorial and gradient-based learning algorithms under a broad range of conditions, up to 1000 variables with an average degree of 20 in Section 5. Finally, to demonstrate its practicality, we apply BOSS to two sets of resting-state fMRI data: simulated data with pseudo-empirical noise distributions derived from randomized empirical fMRI cortical signals, and clinical data from 3T fMRI scans processed into 379 cortical parcels in Section 6. Section 7 provides a brief discussion.

## 1.1 Our Contributions

Our novel contributions to the field include the following:

1. We present a new data structure, Grow-Shrink Trees (GSTs), for efficiently caching results of permutation-based structure learning algorithms. Using GSTs dramatically speeds up many existing permutation-based methods such as GRaSP.

2. We present a new structure learning algorithm, the Best Order Score Search (BOSS), which has similar performance as GRaSP (and thus superior performance to other DAG-learning methods), but is more convenient to use than GRaSP because it has fewer tuning parameters and is faster and more scalable.

3. We prove that BOSS is asymptotically correct and provide an implementation within the TETRAD project[17].

4. We present validations of BOSS's finite sample performance via standard graphical model simulations and on both real and simulated fMRI data.

## 2 Background

The following conventions are used throughout this paper: $V$ denotes a non-empty finite set of variables which double as vertices in the graphical context, lowercase letters $a, b, c, \ldots \in V$ denote variables or singletons, uppercase letters $A, B, C, \ldots \subseteq V$ denote sets, and $X$ denotes a collection of random variables indexed by $V$ whose joint probability distribution is denoted by $P$.

Probabilistic conditional independence between the members of $X$ corresponding to disjoint sets $A, B, C \subseteq V$ is denoted $A \perp\!\!\!\perp B \mid C \, [\, P \,]$ and reads $X_A$ and $X_B$ are independent given $X_C$.

## 2.1 Permutations

A *permutation* is a sequence of variables $\pi = \langle a, b, c, \ldots \rangle$. Let $v \in V$ and $i \in \mathbb{N}$ $(i \leq |V|)$. If $\pi$ is a permutation of $V$, then $\pi$ is equipped with two methods: $\pi.\texttt{index}(v)$ which returns the index of $v$ in $\pi$, and $\pi.\texttt{move}(v, i)$ which returns the permutation obtained by removing $v$ from $\pi$ and reinserted at position $i$. Furthermore, $\text{pre}_\pi(v) \equiv \{w \in V \, : \, \pi.\texttt{index}(w) < \pi.\texttt{index}(v)\}$ is the *prefix* of $v$.

## 2.2 DAG Models

A *directed acyclic graph* (DAG) model is a set probabilistic models whose conditional independence relations are described graphically by a DAG. In general, the vertices and edges of a DAG represent

variables and conditional dependence relations, respectively. We use the notation $\mathcal{G} = (V, E)$ where $\mathcal{G}$ is a graph, $V$ is a vertex set, and $E$ is an edge set. In a DAG, the edge set is comprised of ordered pairs that represent directed edges[1] and contains no directed cycles.

Let $\mathcal{G} = (V, E)$ be a DAG and $v \in V$:

$$\mathrm{pa}_{\mathcal{G}}(v) \equiv \{w \in V \ : \ w \to v \text{ in } \mathcal{G}\}$$
$$\mathrm{ch}_{\mathcal{G}}(v) \equiv \{w \in V \ : \ v \to w \text{ in } \mathcal{G}\}$$

are the *parents* and *children* of $v$, respectively.

Graphical conditional independence between disjoint subsets $A, B, C \subseteq V$ can be read off of a DAG using *d-separation* and is denoted $A \perp\!\!\!\perp B \mid C \, [\mathcal{G}]$. This criterion admits the concept of a *Markov equivalence class* (MEC) which (in the context of this paper) is a collection of DAGs that represent the same conditional independence relations.

## 2.3 Causal Discovery

DAGs are connected to causality by the *causal Markov* and *causal faithfulness* assumptions. A DAG is *causal* if it describes the true underlying causal process by placing a directed edge between a pair of variables if and only if the former directly causes the latter.

*Causal Markov*: If $\mathcal{G} = (V, E)$ is causal for a collection of random variables $X$ indexed by $V$ with probability distribution $P$, then for disjoint subsets $A, B, C \subseteq V$:

$$A \perp\!\!\!\perp B \mid C \, [\mathcal{G}] \quad \Rightarrow \quad A \perp\!\!\!\perp B \mid C \, [P].$$

More generally, this implication is called the *Markov property* and we say DAGs satisfying this property *contains* the distribution. Moreover, a DAG $\mathcal{G}$ is *subgraph minimal*[2] if no subgraph of $\mathcal{G}$ contains $P$.

*Causal faithfulness*: If $\mathcal{G} = (V, E)$ is causal for a collection of random variables $X$ indexed by $V$ with probability distribution $P$, then for disjoint subsets $A, B, C \subseteq V$:

$$A \perp\!\!\!\perp B \mid C \, [P] \quad \Rightarrow \quad A \perp\!\!\!\perp B \mid C \, [\mathcal{G}].$$

Accordingly, under the causal Markov and causal faithfulness assumptions, finding the MEC of the causal DAG is equivalent to identifying the simplest model that contains the data generating distribution. From this point of view, causal discovery is a standard model selection problem which we address using the Bayesian information criterion (BIC) [7, 21].

Let $\mathcal{G}$ be a DAG and $\boldsymbol{X} \stackrel{\text{iid}}{\sim} P$ be a dataset drawn from a member of a curved exponential family:

$$\mathtt{BIC}(\mathcal{G}, \boldsymbol{X}) = \sum_{v \in V} \mathtt{BIC}(\boldsymbol{X}_v, \boldsymbol{X}_{\mathrm{pa}_{\mathcal{G}}(v)})$$
$$= \sum_{v \in V} \ell_{v|\mathrm{pa}_{\mathcal{G}}(v)}(\hat{\theta}_{\mathrm{mle}} \mid \boldsymbol{X}) - \frac{\lambda}{2} |\hat{\theta}_{\mathrm{mle}}| \log(n)$$

where $\lambda > 0$ is a penalty discount, $\ell$ is the log-likelihood function, and $\theta_{\mathrm{mle}}$ is the maximum likelihood estimate of the parameters; for details on curved exponential families, we refer reader to [9]. Importantly, the BIC is *consistent*.

**Proposition 1.** *Haughton [7] If $P$ is a member of a curved exponential family and $\boldsymbol{X} \stackrel{\text{iid}}{\sim} P$, then the BIC is maximized in the large sample limit by DAG models containing $P$ that minimize the dimension of the parameters space.*

Notably, Gaussian and multinomial DAG models form curved exponential families in which the causal DAG minimizes the dimension of the parameters space, and thereby maximize the BIC [11]. For these distributions, DAG models belonging to the same MEC correspond to the same DAG model,

---

[1] The edge $v \leftarrow w$ corresponds to the order pair $(v, w)$.

[2] This concept is also known as SGS-minimality [28].

so it is common for algorithms to return a graphical representation of the MEC called a CPDAG rather than a DAG. Choosing between DAG models in a MEC requires additional information.

We greedily search over permutations of variables using grow-shrink (GS) [13] to project and score DAGs from permutations. Algorithm 1 (`grow`) and Algorithm 2 (`shrink`) give the details of GS while Algorithm 3 (`project`) details the process of projecting a permutation to a DAG.

---

**Algorithm 1:** $\texttt{grow}(\boldsymbol{X}, v, Z)$

**Input:** $\texttt{data}: \boldsymbol{X}$ $\texttt{var}: v$ $\texttt{prefix}: Z$
**Output:** $\texttt{parents}: W$
$W \leftarrow \varnothing$
**repeat**
    $w \leftarrow \texttt{argmax}_{z \in Z} \texttt{BIC}(\boldsymbol{X}_v, \boldsymbol{X}_{W \cup z})$
    **if** $w \neq \varnothing$ **then**
        $W \leftarrow W \cup w$
**until** $w = \varnothing$

---

**Algorithm 2:** $\texttt{shrink}(\boldsymbol{X}, v, W)$

**Input:** $\texttt{data}: \boldsymbol{X}$ $\texttt{var}: v$ $\texttt{parents}: W$
**Output:** $\texttt{parents}: W$
**repeat**
    $w \leftarrow \texttt{argmax}_{w \in W} \texttt{BIC}(\boldsymbol{X}_v, \boldsymbol{X}_{W \setminus w})$
    **if** $w \neq \varnothing$ **then**
        $W \leftarrow W \setminus w$
**until** $w = \varnothing$

---

**Algorithm 3:** $\texttt{project}(\boldsymbol{X}, \pi)$

**Input:** $\texttt{data}: \boldsymbol{X}$ $\texttt{perm}: \pi$
**Output:** $\texttt{DAG}: \mathcal{G}$
$E \leftarrow \varnothing$
**foreach** $v \in \pi$ **do**
    $Z \leftarrow \texttt{pre}_\pi(v)$
    $W \leftarrow \texttt{grow}(\boldsymbol{X}, v, Z)$
    $W \leftarrow \texttt{shrink}(\boldsymbol{X}, v, W)$
    **foreach** $w \in W$ **do**
        $E \leftarrow E \cup (v, w)$
$\mathcal{G} \leftarrow (V, E)$

---

However, how do we know that the causal DAG will even be constructed by `project`? This is guaranteed by the follow results. Let $P$ is a member of a curved exponential family satisfying causal Markov and causal faithfulness:

- If $\mathcal{G}$ is the causal DAG, then $\mathcal{G}$ is subgraph minimal [11, 28].

- If $\mathcal{G}$ is a DAG and $\boldsymbol{X} \overset{\text{iid}}{\sim} P$ then $\mathcal{G}$ is subgraph minimal if and only if there exists a permutation $\pi$ such that $\mathcal{G} = \texttt{project}(\boldsymbol{X}, \pi)$ in the large sample limit [12].

These results admit the following strategy: (1) Search over permutations while using GS and the BIC to construct and score subgraph minimal DAG from these permutations. (2) Return the MEC of the subgraph minimal DAG that maximizes the BIC. The question is how to do so in a way that is both comprehensive and efficient.

## 3 Grow-Shrink Trees

Grow-shrink trees (GSTs) are tree data structures for caching the results of the `grow` and `shrink` subroutines of GS [13]. This data structure is compatible with many permutation-based structure learning algorithms, including BOSS and GRaSP. As an initialization step, a GST is constructed for each variable in the dataset which are then queried during search rather than running GS. Each tree has a root node representing the empty parent set. Parents are accumulated by traversing edges of the tree, with every node in the tree corresponding to a "grown" parent set. Each grown parent set corresponds to running the `grow` subroutine for some prefix. The `shrink` subroutine is also run and cached at each node of the tree.

The benefit of using GSTs is in how efficiently they store information needed for running GS. In the `grow` subroutine, nodes are added to the GST one at a time and scored. A new child node is added to the tree for each possible addition, and the child nodes are sorted in descending order relative to their scores. After all candidate parents have been added to the tree, the first child in the sorted list that is also in the prefix is chosen and the corresponding edge traversed. The `shrink` subroutine can then be run and the removed parents and scores can be cached.

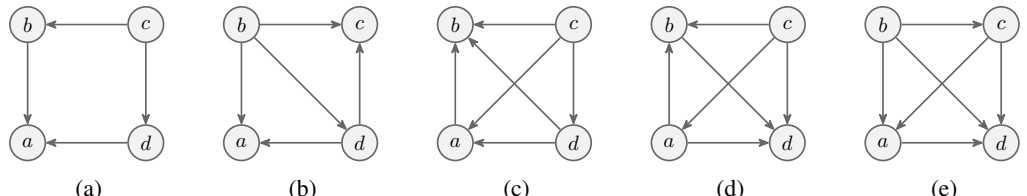

Figure 1: Minimal subgraphs.

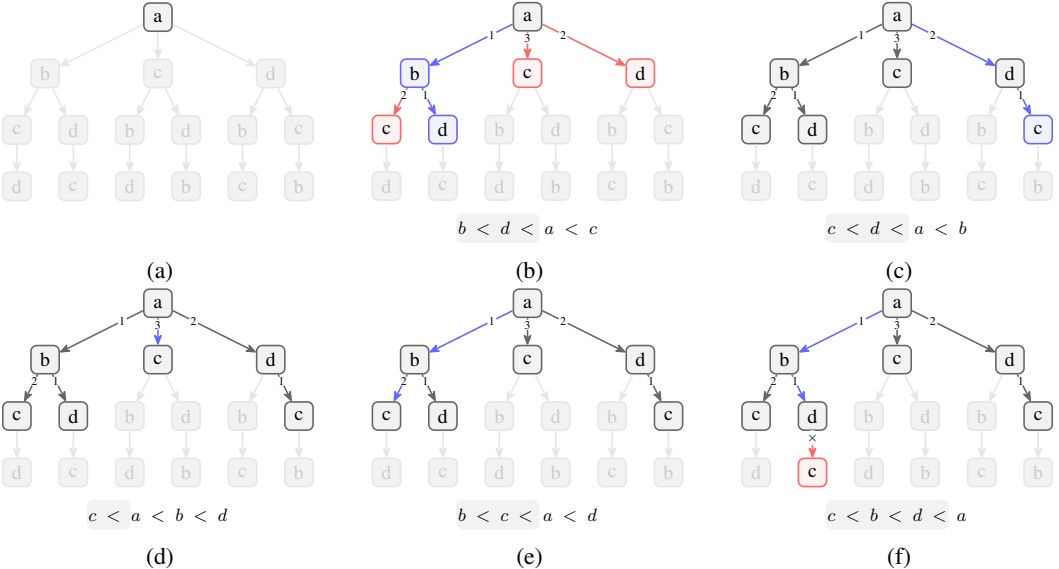

Figure 2: A Grow-shrink Tree

## 3.1 An Example

This section walks through the usage of a GST applied to a toy example. Minimal subgraphs are depicted in Figure 1 of which (1a) is the true DAG. Figure 2 outlines the process of growing a GST for $a$ while scoring permutations. In the following, we identify nodes in the GST by their path from the root. All ambiguities in sorting the branches of a node are resolved in alphabetical order. We proceed in sequence, but in theory some of these steps could be performed in parallel.

In what follows we run GS on a series of permutations while caching the results in a GST. We describe the nodes of the tree by the sequence of variables traversed to reach them from the root. We use colors to track our progress: light gray parts of the tree denote unexplored paths with no cached scores, dark gray parts of the tree denote explored paths with cached scores. Colored edges denote the paths considered during the execution of GS: red for rejected and blue for accepted. Colored nodes denote the same, but are only colored if they do not have a cached value and require a new score calculation. Numbered edges directed out of a parent node record the order of preference for the children from high to low. To simplify this example, we do not cache the results of shrink. In (2a) we initialize the tree by adding the root node $a$.

Our first permutation to evaluate is $\langle b, d, a, c \rangle$ which is shown in (2b) and whose minimal subgraph is depicted in (1b). Node $a$ has not been expanded, so we score and sort nodes $ab$, $ac$, and $ad$. Node $ab$ scored the highest and is contained in the prefix, so we travel to node $ab$. Node $ab$ has not been expanded, so we score and sort nodes $abc$ and $abd$. Node $abd$ scored the highest and is contained in the prefix, so we travel to node $abd$. At this point no more variables are available in the prefix so we have completed growing. We run shrink at node $abd$ which removes nothing so the GS result is $\{b, d\}$.

Our next permutation to evaluate is $\langle c, d, a, b \rangle$ which is shown in (2c) and whose minimal subgraph is depicted in (1c). Node $a$ has already been expanded so we check its outgoing edges. Node $ab$

has the highest score but is not contained in the prefix. Node $ad$ has the second highest score and is contained in the prefix, so we travel to node $ad$. Node $ad$ has not been expanded, so we score and sort nodes $adc$. Node $c$ has the highest score and is contained in the prefix, so we travel to node $adc$. At this point no more variables are available in the prefix so we have completed growing. We run shrink at node $adc$ which removes nothing so the GS result is $\{c, d\}$.

Our next permutation to evaluate is $\langle c, a, b, d \rangle$ which is shown in (2d) whose minimal subgraph is depicted in (1d). Node $a$ has already been expanded so we check its outgoing edges. Node $ab$ has the highest score but is not contained in the prefix. Node $ad$ has the second highest score but is not contained in the prefix. Node $ac$ has the third highest score and is contained in the prefix, so we travel to node $ac$. At this point no more variables are available in the prefix so we have completed growing. We run shrink at node $ac$ which removes nothing so the GS result is $\{c\}$.

Our next permutation to evaluate is $\langle b, c, a, d \rangle$ which is shown in (2e) whose minimal subgraph is depicted in (1e). Node $a$ has already been expanded so we check its outgoing edges. Node $ab$ has the highest score and is contained in the prefix, so we travel to node $ab$. $ab$ has already been expanded so we check its outgoing edges. Node $abd$ has the highest score but is not contained in the prefix. Node $abc$ has the second highest score and is contained in the prefix, so we travel to node $abc$. At this point no more variables are available in the prefix so we have completed growing. We run shrink at node $abc$ which removes nothing so the GS result is $\{b, c\}$.

Our last permutation to evaluate is $\langle c, b, d, a \rangle$ which is shown in (2f) and whose minimal subgraph is depicted in (1a). Node $a$ has already been expanded so we check its outgoing edges. Node $ab$ has the highest score and is contained in the prefix, so we travel to node $ab$. Node $ab$ has already been expanded so we check its outgoing edges. Node $d$ has the highest score and is contained in the prefix, so we travel to node $abd$. Node $abd$ has not been expanded, so we score node $abdc$. No scores result in an improvement to the overall score so we have completed growing. We run shrink at node $abd$ which removes nothing so the GS result is $\{b, d\}$.

## 4 Best Order Score Search

Similar to GES, BOSS uses a two phase search procedure [5]. The first phase uses the `best-move` method, which takes a variable as input and greedily moves it to the position in the current permutation that maximizes the score. In this phase, `best-move` is repeatedly applied to each variable, one at a time, until there are no more moves that increase the score. This phase concludes with the `find-compelled` procedure of [4] which converts the DAG into a CPDAG. The second phase of BOSS is BES which is exactly second phase of GES. The BES step is optional but guarantees asymptotic correctness if executed.

---

**Algorithm 4:** $\texttt{BOSS}(\boldsymbol{X}, \pi, \delta)$

**Input:** data : $\boldsymbol{X}$ perm : $\pi$ flag : $\delta$
**Output:** graph : $\mathcal{G}$
$\mathcal{T} \leftarrow \texttt{GST}(\boldsymbol{X})$
**repeat**
    best $\leftarrow \mathcal{T}.\texttt{score}(\pi)$
    **foreach** $v \in \pi$ **do**
        $\pi \leftarrow \texttt{best-move}(\mathcal{T}, \pi, v)$
**until** best $= \mathcal{T}.\texttt{score}(\pi)$
$\mathcal{G} \leftarrow \mathcal{T}.\texttt{project}(\pi)$
$\mathcal{G} \leftarrow \texttt{find-compelled}(\mathcal{G})$
**if** $\delta = \texttt{true}$ **then**
    $\mathcal{G} \leftarrow \texttt{BES}(\mathcal{G}, \boldsymbol{X})$

---

**Algorithm 5:** $\texttt{best-move}(\mathcal{T}, \pi, v)$

**Input:** GSTs : $\mathcal{T}$ perm : $\pi$ var : $v$
**Output:** perm : $\pi$
best $\leftarrow \mathcal{T}.\texttt{score}(\pi)$
**for** $i \leftarrow 1$ **to** $|\pi|$ **do**
    $j \leftarrow \pi.\texttt{index}(v)$
    $\pi \leftarrow \pi.\texttt{move}(v, i)$
    **if** best $< \mathcal{T}.\texttt{score}(\pi)$ **then**
        best $\leftarrow \mathcal{T}.\texttt{score}(\pi)$
    **else**
        $\pi \leftarrow \pi.\texttt{move}(v, j)$

---

**Proposition 2.** *Let $P$ be a member of a curved exponential family satisfying causal Markov and causal faithfulness. If $\boldsymbol{X} \overset{\text{iid}}{\sim} P$ then* $\texttt{BOSS}(\boldsymbol{X}, \pi, \texttt{true})$ *returns the MEC of the causal DAG for all initial permutation $\pi$ in the large sample limit.*

*Proof.* Since `project` returns a subgraph minimal DAG, it contains $P$. Accordingly, asymptotic correctness follows from the correctness of BES. $\qquad\square$

In Algorithm 4 (`BOSS`) and Algorithm 5, `GST` constructs a collection of GSTs, denoted $\mathcal{T}$, which contains one GST for each variable. The collection $\mathcal{T}$ is equipped with `project` and `score` methods which runs the GS algorithm to project and score a permutation, respectively.

## 5 Simulations

We evaluated the speed and performance of `BOSS` on simulated data compared to other algorithms: `GRaSP`, `fGES`, `PC`, `DAGMA`, and `LiNGAM`. Our evaluation used the performance metrics tabulated in Table 1 which were originally proposed by [10]. All algorithms were run on an Apple M1 Pro processor with 16G of RAM. The results reported in the main text are abridged but the complete results are available in the Supplement.

Table 1: Metrics

| True | Estimated | Adjacency | Orientation |
|---|---|---|---|
| | $a \leftarrow b$ | `tp` | `tp, tn` |
| | $a \rightarrow b$ | `tp` | `fp, fn` |
| $a \leftarrow b$ | $a - b$ | `tp` | `fn` |
| | $a \ldots b$ | `fn` | `fn` |
| | $a \leftarrow b$ | `fp` | `fp` |
| | $a \rightarrow b$ | `fp` | `fp` |
| $a \ldots b$ | $a - b$ | `fp` | |
| | $a \ldots b$ | `tn` | |

$$\text{Precision} = \frac{\texttt{tp}}{\texttt{tp} + \texttt{fp}}$$

$$\text{Recall} = \frac{\texttt{tp}}{\texttt{tp} + \texttt{fn}}$$

We evaluated our implementation of `BOSS` using a BIC score with $\lambda = 2$ and no BES step as it did not appear to improve performance. For `GRaSP`, we modified (to use GSTs) and used the TETRAD implementation with the same parameters as the authors and a BIC score with $\lambda = 2$. [12, 17]. For `fGES` we used the TETRAD implementation with default parameters and a BIC score with $\lambda = 2$ [5, 15, 17]. We also used the implementation of `PC` in TETRAD with default parameters using a BIC score with $\lambda = 2$ as a conditional independence oracle [17, 25]. For `DAGMA` we used the authors' Python implementation with the parameters reported in their paper but changed the threshold to 0.1 [2] and used the technique described in [14] to resolve cycles. Moreover, the output of `DAGMA` was converted to a CPDAG for linear Gaussian simulations. For `LiNGAM` we used the authors' Python implementation with default parameters [8, 22].

We generated Erdős-Rényi DAGs by applying an arbitrary order to vertices of an Erdős-Rényi graph. For scale-free networks, first we generated an Erdős-Rényi DAG and then redrew the parents of each vertex according to the Barabási-Albert model [1]. This resamples the out-degrees distribution to be scale-free while keeping in-degree distribution constant. This procedure was motivated by the results in Figure 5. Edge weights were sampled from a uniform distribution in the interval [-1.0, 1.0] and error distributions were generated with standard deviations sampled from a uniform distribution in the interval [1.0, 2.0]. The complete simulation details, including figures plotting the simulated scale-free in/out degree distributions, are include in the Supplement. Additionally, the data are available for download at: `https://github.com/cmu-phil/boss`.

Figure 3 compares `BOSS` against `GRaSP`, `fGES`, `PC`, and `DAGMA` on linear Gaussian data generated from (3a) Erdős-Rényi and (3b) scale-free networks. The output of `DAGMA` was converted to a CPDAG for these simulations. Lam et al. [12] attribute the excellent performance of `GRaSP` to it being robust against the ubiquity of almost-violations of causal faithfulness [29, 27]. Due to the algorithmic and performance parallels between `GRaSP` and `BOSS`, we conjecture that a similar argument could be made for `BOSS`.

Figure 4a compares `BOSS` against `DAGMA` and `LiNGAM` on linear exponential and linear Gumbel data from scale-free networks. Interestingly, `LiNGAM`, which takes advantage of non-Gaussian signal in data, does not perform appreciably better than `BOSS`.

Figure 4b compares `BOSS` against `GRaSP` on linear Gaussian data generated from scale-free networks with a focus on scalability; runs exceeding two hours were cancelled and are not reported. Here we see that `BOSS` maintains a high level of accuracy while scaling much better than `GRaSP`. It is also

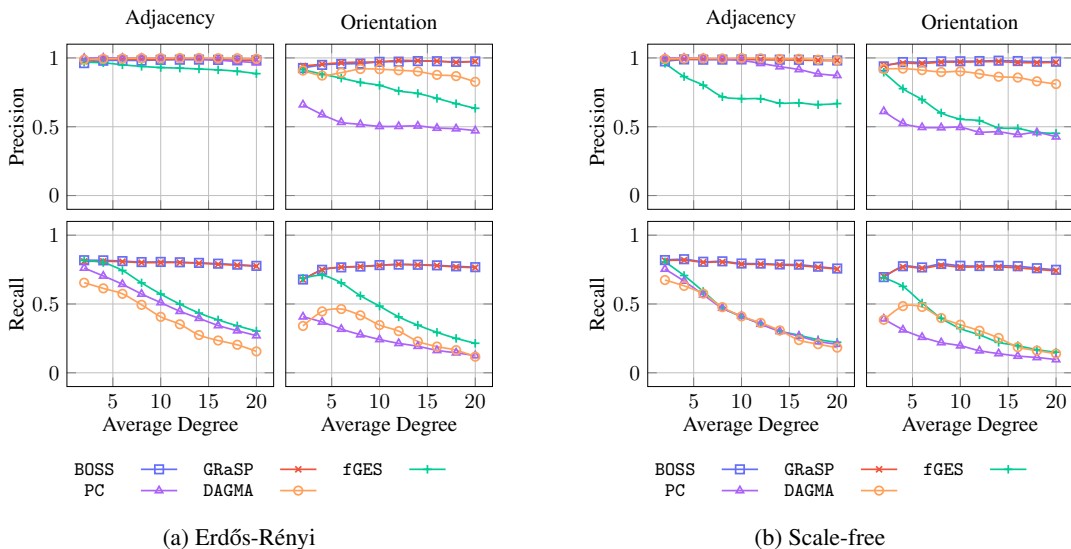

(a) Erdős-Rényi

(b) Scale-free

Figure 3: Mean statistics over 20 repetitions: 100 variables and sample size 1,000.

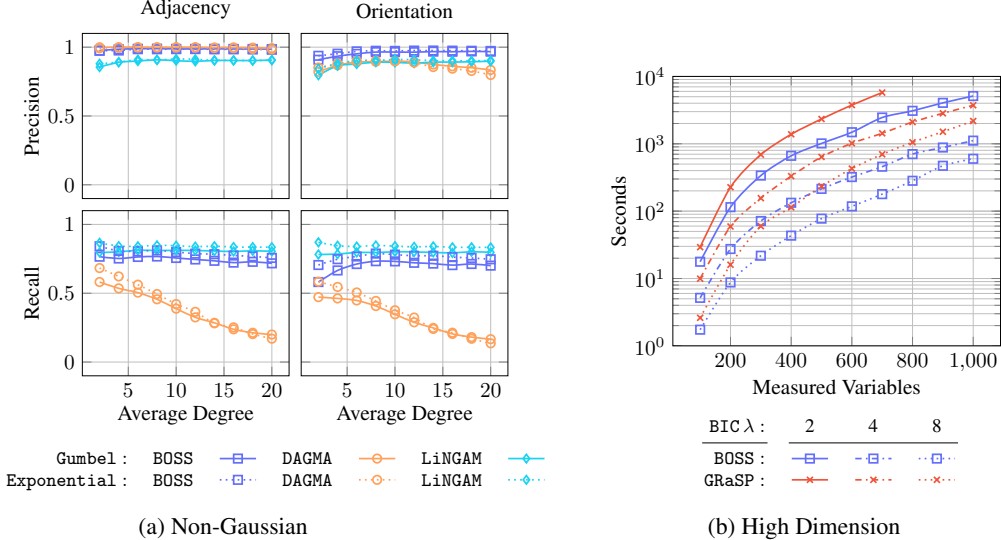

(a) Non-Gaussian

(b) High Dimension

Figure 4: Mean statistics over 20 repetitions: scale-free, 100 variables, average degree 20 (when not varied), and sample size 1,000.

important to note that nearly all of these simulations were infeasible for `GRaSP` before implementing it with GSTs. Table 2, the corresponding table, only reports results for `BOSS` since the two algorithms have nearly identical performance on all statistics except for running time. Full results are tabulated in the Supplement.

## 6  Validation on fMRI

The high spatial coverage of fMRI's has allowed researchers to study brain function at different scales, from voxels to cortical parcellations to functional systems (such as default mode or visual systems). Given the potential differences in spatial dimension, the expected connectivity density of functional brain networks remains unclear. For example, even after keeping only the 75th percentile stronger connections, structural connectivity networks from 90 cortical regions [23] had 1012 connections on average. Since structural connectivity supports functional connectivity, we could expect empirical

Table 2: Mean statistics over 10 repetitions: scale-free, average degree 20, and sample size 1,000.

| Variables | Algorithm | BIC $\lambda$ | Adj Pre | Adj Rec | Ori Pre | Ori Rec |
|---|---|---|---|---|---|---|
| | | 2 | 0.98 | 0.80 | 0.97 | 0.80 |
| 500 | BOSS | 4 | 1.00 | 0.72 | 0.99 | 0.72 |
| | | 8 | 1.00 | 0.57 | 0.99 | 0.57 |
| | | 2 | 0.97 | 0.80 | 0.97 | 0.80 |
| 1000 | BOSS | 4 | 1.00 | 0.73 | 1.00 | 0.72 |
| | | 8 | 1.00 | 0.59 | 1.00 | 0.58 |

Table 3: fMRI simulated data with pseudo-empirical errors

| Algorithm | Adj Pre | Adj Rec | Ori Pre | Ori Rec | $\Delta$BIC | Edges | Seconds |
|---|---|---|---|---|---|---|---|
| BOSS | 0.99 | 0.94 | 0.96 | 0.90 | 211.79 | 951.40 | 15.46 |
| fGES | 0.97 | 0.60 | 0.70 | 0.43 | 7784.48 | 617.35 | 5.16 |
| DAGMA | 1.00 | 0.69 | 0.98 | 0.67 | 3080.85 | 687.15 | 54.58 |
| LiNGAM | 0.54 | 0.94 | 0.35 | 0.62 | 3868.93 | 1752.75 | 582.03 |

functional connectivity networks to be in that order of magnitude. Previous studies applying causal discovery methods to fMRI simulated networks with a high number of variables and connections have shown that while the adjacency recovery precision of these methods can be very high, they usually show a low adjacency recall [16, 20]. Despite possibly having low recall, the limited applications of causal discovery methods to real world data [3, 18] often recover models with average degree greater than 20. Considering this limitation and the likelihood that real fMRI brain networks have high average connectivity (degree), causal discovery methods capable of reducing the number of false-negative connections will substantially improve future analysis of fMRI data.

To demonstrate its practical utility in this important real-world domain, we apply BOSS to two types of resting-state fMRI data: simulated data with pseudo-empirical noise distributions derived from randomized empirical fMRI cortical signals and clinical data from 3T fMRI scans processed into cortical parcels.

## 6.1 Simulated fMRI

We simulated fMRI data following the approach in [19]. Networks were based on a directed random graphical model that prefers common causes and causal chains over colliders. 40 networks were simulated with 200 variables and an average degree of 10. Edge weights were sampled from a uniform distribution in the interval $[0.1, 0.4]$, randomly setting $10\%$ of the coefficients to their negative value. Pseudo-empirical noise terms were produced by randomizing fMRI resting-state data across data points, regions, and participants from the Human Connectome Project (HCP). Using pseudo-empirical terms better captures the marginal distributional properties of the empirical fMRI. One thousand data points were generated from this procedure. These data are available for download at: https://github.com/cmu-phil/boss.

Accuracy and timing results are shown in Table 3 for BOSS, fGES, DAGMA, and LiNGAM. These results show that BOSS has by far the best BIC score of the group, compared to the BIC score of the true model, and that the running time of BOSS is very reasonable for a problem of this size. Precisions and recalls for BOSS are quite high, for both adjacencies and orientations. The poor performance of LiNGAM is due to insufficient non-Gaussian signal. More details are included in the Supplement.

## 6.2 Clinical fMRI

We applied BOSS to 171 3-Tesla resting state fMRI scans from patients beginning treatment for alcohol use disorder [3]. Before applying BOSS, the data were cleaned and parcellated into 379 variables representing biologically interpretable and spatially contiguous regions [6]. More details on the data collection, cleaning, and processing can be found in [3]. Figure 5 reports the in/out-degree distributions of the resulting models. The solid line depicts the median degree across all graphs and

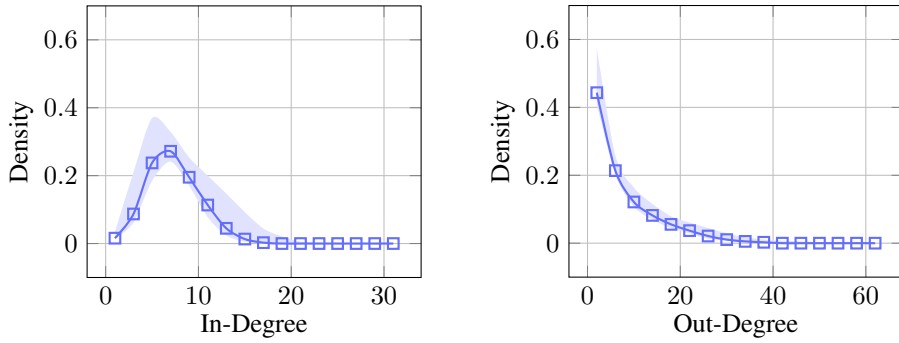

Figure 5: In/out-degree distributions on clinical resting state fMRI data.

the color region shades in the area between the 2.5 and 97.5 percentiles. These plots indicate that the models are scale-free, with a small number of hub vertices with much higher degree than other vertices. Scale-free connectivity is consistent with biological expectations and prior connectivity analysis on fMRI data [18]. More details are included in the Supplement.

# 7 Discussion

We have proposed a successor to the GRaSP algorithm [12] that retains its high accuracy but is faster and more scalable. BOSS implemented with GSTs comfortably scales to least 1000 variables with an average degree of at least 20. It can comfortably and informatively analyze data from 400 or even 1000 densely connected fMRI cortical parcellations. We show in simulations that our method is highly accurate for Erdős-Rényi graphs as well as scale-free graphs. Despite being developed for the linear Gaussian case, BOSS also performs well in the linear non-Gaussian case. BOSS is fast and accurate on simulated fMRI data and can rapidly produce informative and plausible models from clinical fMRI data. BOSS is available for use within the TETRAD project which includes Python and R wrappers [17].

The success of BOSS presents an opportunity to pursue further theoretical work showing how BOSS differs from GRaSP and what general lessons we may learn for constructing successor algorithms to BOSS. There is substantial room for additional improvements, as BOSS does have several limitations. For example, BOSS cannot handle most forms of unmeasured confounding, so it will be valuable to explore ways of adding this functionality while maintaining its accuracy and scalability. It will also be informative to apply BOSS to other types of data such as functional genomic data, financial data, and electronic health records.

## Acknowledgments and Disclosure of Funding

We thank our anonymous reviewers for their detailed and insightful comments. BA was supported by the US National Institutes of Health under the Comorbidity: Substance Use Disorders and Other Psychiatric Conditions Training Program T32DA037183. JR was supported by the US Department of Defense under Contract Number FA8702-15-D-0002 with Carnegie Mellon University for the operation of the Software Engineering Institute. RSR was supported by the US National Institutes of Health under awards R01AG055556 and R01MH109520. JC was supported by the US National Institutes of Health under awards K01AA026349 and UL1TR002494, the UMN Medical Discovery Team on Addiction, and the Westlake Wells Foundation. EK was supported by the US National Institutes of Health under award UL1TR002494. The content of this paper is solely the responsibility of the authors and does not necessarily represent the official views of these funding agencies. All authors have no conflicts of interest to report.

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
