# Supplement for Fast Scalable and Accurate Discovery of DAGs Using the Best Order Score Search and Grow-Shrink Trees

**Bryan Andrews**
Department of Psychiatry & Behavioral Sciences
University of Minnesota
Minneapolis, MN 55454
andr1017@umn.edu

**Joseph Ramsey**
Department of Philosophy
Carnegie Mellon University
Pittsburgh, PA 15213
jdramsey@andrew.cmu.edu

**Rubén Sánchez-Romero**
Center for Molecular and Behavioral Neuroscience
Rutgers University
Newark, NJ 07102
ruben.saro@rutgers.edu

**Jazmin Camchong**
Department of Psychiatry & Behavioral Sciences
University of Minnesota
Minneapolis, MN 55454
camch002@umn.edu

**Erich Kummerfeld**
Institute for Health Informatics
University of Minnesota
Minneapolis, MN 55454
erichk@umn.edu

## 1 Simulations

Linear Gaussian simulations are included to establish the performance of BOSS compared to existing causal discovery algorithms on Erdős-Rényi and scale-free graphs. These results are tabulated in Tables 1, 2, 3, and 4 which compare the following algorithms:

- BOSS
- GRaSP
- fGES
- PC
- DAGMA - converted to CPDAG

LiNGAM was not included in the linear Gaussian comparison because no non-Gaussian signal is available. Linear Non-Gaussian simulations are included to establish the performance of BOSS compared to methods that (can) take advantage of non-Gaussian signal. These results are tabulated in Tables 5, 6, 7, and 8 which compare the following algorithms:

- BOSS
- DAGMA
- LiNGAM

High dimensional simulations are included to establish the scalability of BOSS compared to GRaSP. These results are tabulated in Tables 9 and 10.

37th Conference on Neural Information Processing Systems (NeurIPS 2023).

## 1.1 Algorithms

In what follows, we give a complete list of the tuning parameter settings we used for all algorithms:

- BOSS - Best Order Score Search
    - SemBicScore()
        * setPenaltyDiscount(2/4/8)
        * setStructurePrior(0)
        * setRuleType(SemBicScore.RuleType.CHICKERING)
    - setUseBes(False)
    - setNumStarts(1)
    - setNumThreads(1)
    - setUseDataOrder(false)
- GRaSP - Greedy Relaxations of the Sparest Permutation [5, 7]
    - SemBicScore()
        * setPenaltyDiscount(2/4/8)
        * setStructurePrior(0)
        * setRuleType(SemBicScore.RuleType.CHICKERING)
    - setDepth(3)
    - setNonSingularDepth(1)
    - setUncoveredDepth(1)
    - setNumStarts(1)
    - setOrdered(false)
    - setUseDataOrder(false)
- fGES - fast Greedy Equivalent Search [3, 6, 7]
    - SemBicScore()
        * setPenaltyDiscount(2)
        * setStructurePrior(0)
        * setRuleType(SemBicScore.RuleType.CHICKERING)
    - setFaithfulnessAssumed(false)
    - setMaxDegree($-1$)
- PC - Peter and Clark [7, 9]
    - ScoreIndTest(SemBicScore())
        * setPenaltyDiscount(2)
        * setStructurePrior(0)
        * setRuleType(SemBicScore.RuleType.CHICKERING)
    - setDepth(1000)
    - setStable(false)
    - setConflictRule(ConflictRule.OVERWRITE_EXISTING)
    - setAggressivelyPreventCycles(false)
- DAGMA - DAGs via M-matrices for Acyclicity [2]
    - loss_type $=' l2'$
    - lambda1 $= 0.1$
    - w_threshold $= 0.1$
    - T $= 4$
    - mu_init $= 1.0$
    - mu_factor $= 0.1$
    - s $= [1.0, .9, .8, .7]$
    - warm_iter $= 2e4$
    - max_iter $= 7e4$
    - lr $= 0.0003$

- – `checkpoint` = 1000
- – `beta_1` = 0.99
- – `beta_2` = 0.999

- • `LiNGAM` - Linear Non-Gaussian Acyclic Model [4, 8]
  - – `random_state` = None
  - – `prior_knowledge` = None
  - – `apply_prior_knowledge_softly` = False
  - – `measure` $='$ `pwling`$'$

## 1.2 Data-generation

Erdős-Rényi and scale-free graphs were generated according to Algorithms 2 and 3, respectively. For scale-free graphs, the parents of an Erdős-Rényi graph are redrawn to produce a graph whose out-degree follows a power-law. The redrawing process follows a modified Barabási-Albert model where the preferential attachment of each potential parent is inflated by one to account for vertices with zero out-degree [1]. Figure 1 depicts the in/out degree distributions for scale-free graphs (100 vertices, average degree 10) where the solid black line gives the median and the grayed region denotes an empirical 95% confidence interval. Notably, the in/out-degree for Erdős-Rényi graphs will both follow the in-degree distribution defined in Figure 1.

Data were generated according to Algorithm 1 where $\mathcal{U}$ denotes a uniform distribution. In our simulations, the error $\mathcal{E}$ is either distributed as Gaussian, Gumbel, or Exponential. In all cases, the columns of the data matrix were shuffled prior to being passed to a search algorithm, so that the variable order in the dataset does not match the order of data generation.

---

**Algorithm 1:** `simulate`$(\mathcal{G}, \mathcal{E}, n)$

---

**Input:** `DAG` : $\mathcal{G} = (V, E)$ `error` : $\mathcal{E}$ `samples` : $n$
**Output:** `data` : $\boldsymbol{X}$
**foreach** $v \in V$ **do**

    $\sigma \sim \mathcal{U}(1, 2)$
    **foreach** $w \in \mathrm{pa}_{\mathcal{G}}(v)$ **do**
        $\beta \sim \mathcal{U}(-1, 1)$
        **foreach** $i \in [n]$ **do**
            $\boldsymbol{X}_{i,v} \sim \mathcal{E}(\sigma)$
            $\boldsymbol{X}_{i,v} \leftarrow \boldsymbol{X}_{i,v} + \beta \boldsymbol{X}_{i,w}$
    $\boldsymbol{X}_v \leftarrow$ `standardize`$(\boldsymbol{X}_v)$

---

---

**Algorithm 2:** `ER-DAG`$(V, \pi, \alpha)$

---

**Input:** `vertices` : $V$ `perm` : $\pi$ `avg deg` : $\alpha$
**Output:** `DAG` : $\mathcal{G} = (V, E)$
$E \leftarrow \varnothing$
**repeat**

    $(v, w) \sim f : V \times V \rightarrow \mathbb{R}$   s.t.   $f \propto 1_{\mathrm{pre}_\pi(v)}(w)$
    $E \leftarrow E \cup \{(v, w)\}$
**until** $|E| = \frac{\alpha}{2} |V|$

---

**Algorithm 3:** $\texttt{SF-DAG}(V, \pi, \alpha)$

---

**Input:** vertices $: V$ perm $: \pi$ avg deg $: \alpha$
**Output:** DAG $: \mathcal{G} = (V, E)$
$\mathcal{H} \leftarrow \texttt{ER-DAG}(V, \pi, \alpha)$
$E \leftarrow \varnothing$
**for** $v \in \pi$ **do**
   **repeat**
      $w \sim f : V \to \mathbb{R}$   s.t.   $f \propto 1_{\mathrm{pre}_\pi(v)}(w) + |\mathrm{ch}_\mathcal{G}(w)|$
      $E \leftarrow E \cup \{(v, w)\}$
   **until** $|\mathrm{pa}_\mathcal{G}(v)| = |\mathrm{pa}_\mathcal{H}(v)|$

---

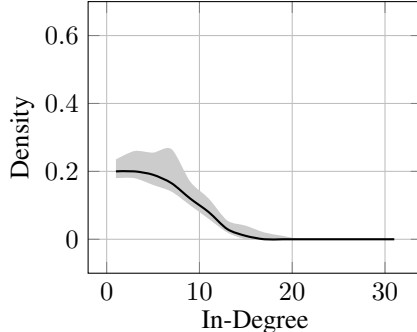 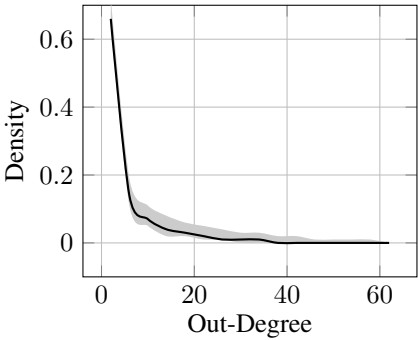

Figure 1: In/out-degree distributions for scale-free simulations with 100 variables and average degree 10 over 100 repetitions.

## 2 Complete Results

Tables 1 and 2 tabulate results for $\texttt{BOSS}$, $\texttt{GRaSP}$, $\texttt{fGES}$, $\texttt{PC}$, and $\texttt{DAGMA}$ run on linear Gaussian models with Erdős-Rényi graphs, 100 variables, and sample size 1000, for average degrees 2 through 20. Table 1 shows adjacency precision, adjacency recall, orientation precision, and orientation recall. Table 2 shows the BIC score difference compared to the true graph, number of edges in the estimated graph, and elapsed wall time in seconds. Mean statistics over 20 repetitions are shown with standard deviations in parentheses.

Tables 3 and 4 tabulate results for $\texttt{BOSS}$, $\texttt{GRaSP}$, $\texttt{fGES}$, $\texttt{PC}$, and $\texttt{DAGMA}$ run on linear Gaussian models with scale-free graphs, 100 variables, and sample size 1000, for average degrees 2 through 20. Table 3 shows adjacency precision, adjacency recall, orientation precision, and orientation recall. Table 4 shows the BIC score difference compared to the true graph, number of edges in the estimated graph, and elapsed wall time in seconds. Mean statistics over 20 repetitions are shown with standard deviations in parentheses.

Tables 5 and 6 tabulate results for $\texttt{BOSS}$, $\texttt{DAGMA}$, and $\texttt{LiNGAM}$ run on linear Gumbel models with scale-free graphs, 100 variables, and sample size 1000, for average degrees 2 through 20. Table 5 shows adjacency precision, adjacency recall, orientation precision, and orientation recall. Table 6 shows the BIC score difference compared to the true graph, number of edges in the estimated graph, and elapsed wall time in seconds. Mean statistics over 20 repetitions are shown with standard deviations in parentheses.

Tables 7 and 8 tabulate results for $\texttt{BOSS}$, $\texttt{DAGMA}$, and $\texttt{LiNGAM}$ run on linear Exponential models with scale-free graphs, 100 variables, and sample size 1000, for average degrees 2 through 20. Table 7 shows adjacency precision, adjacency recall, orientation precision, and orientation recall. Table 8 shows the BIC score difference compared to the true graph, number of edges in the estimated graph, and elapsed wall time in seconds. Mean statistics over 20 repetitions are shown with standard deviations in parentheses.

Tables 9, 10, 11 and 12 tabulate results for $\texttt{BOSS}$ and $\texttt{GRaSP}$ run on linear Gaussian models with scale-free graphs, average degree 20, and sample size 1000, for numbers of variables 100 through

1000; runs exceeding two hours were cancelled and are not reported. Tables 9 and 10 show adjacency precision, adjacency recall, orientation precision, and orientation recall. Tables 11 and 12 show the BIC score difference compared to the true graph, number of edges in the estimated graph, and elapsed wall time in seconds. Mean statistics over 20 repetitions are shown with standard deviations in parentheses.

Tables 13 and 14 tabulate results for `BOSS`, `DAGMA`, `fGES`, and `LiNGAM` run on simulated fMRI data with linear connection functions and pseudo-empirical noise distributions, 200 variables, average degree 10, and sample size 1000. Table 13 shows adjacency precision, adjacency recall, orientation precision, and orientation recall. Table 14 shows the BIC score difference compared to the true graph, number of edges in the estimated graph, and elapsed wall time in seconds. Mean statistics over 20 repetitions are shown with standard deviations in parentheses.

Tables 15 and 16 tabulate results for `BOSS` and `fGES` run on the empirical fMRI data described in the main text. Table 15 shows BIC scores, number of undirected edges, and total number of edges. Table 16 shows the improvement in BIC score and change in number of edges when comparing `BOSS` to `fGES`. Mean statistics over 171 scans are shown with standard deviations in parentheses.

# References

[1] Barabási, A.-L. and Albert, R. (1999). Emergence of scaling in random networks. *science*, 286:509–512.

[2] Bello, K., Aragam, B., and Ravikumar, P. (2022). DAGMA: Learning DAGs via m-matrices and a log-determinant acyclicity characterization. In Koyejo, S., Mohamed, S., Agarwal, A., Belgrave, D., Cho, K., and Oh, A., editors, *Advances in Neural Information Processing Systems*, volume 35, pages 8226–8239. Curran Associates, Inc.

[3] Chickering, D. M. (2002). Optimal structure identification with greedy search. *Journal of machine learning research*, 3:507–554.

[4] Ikeuchi, T., Ide, M., Zeng, Y., Maeda, T. N., and Shimizu, S. (2023). Python package for causal discovery based on LiNGAM. *Journal of Machine Learning Research*, 24:1–8.

[5] Lam, W.-Y., Andrews, B., and Ramsey, J. (2022). Greedy relaxations of the sparsest permutation algorithm. In *Proceedings of the Conference on Uncertainty in Artificial Intelligence*, pages 1052–1062. PMLR.

[6] Ramsey, J., Glymour, M., Sánchez-Romero, R., and Glymour, C. (2017). A million variables and more: the fast greedy equivalence search algorithm for learning high-dimensional graphical causal models, with an application to functional magnetic resonance images. *International journal of data science and analytics*, 3:121–129.

[7] Ramsey, J. D., Zhang, K., Glymour, M., Romero, R. S., Huang, B., Ebert-Uphoff, I., Samarasinghe, S., Barnes, E. A., and Glymour, C. (2018). Tetrad—a toolbox for causal discovery. In *8th international workshop on climate informatics*.

[8] Shimizu, S., Inazumi, T., Sogawa, Y., Hyvarinen, A., Kawahara, Y., Washio, T., Hoyer, P. O., Bollen, K., and Hoyer, P. (2011). DirectLiNGAM: A direct method for learning a linear non-Gaussian structural equation model. *Journal of Machine Learning Research-JMLR*, 12:1225–1248.

[9] Spirtes, P., Glymour, C., and Scheines, R. (2000). *Causation, prediction, and search*. MIT press.

Table 1: Erdős-Rényi - Mean (Standard Deviation) - 20 Repetitions

| Avg Deg | Algorithm | Adj Pre | Adj Rec | Ori Pre | Ori Rec |
|---|---|---|---|---|---|
| 2 | BOSS | 0.96 (0.021) | 0.82 (0.033) | 0.93 (0.044) | 0.68 (0.062) |
| | GRaSP | 0.98 (0.014) | 0.81 (0.031) | 0.94 (0.047) | 0.68 (0.070) |
| | fGES | 0.97 (0.019) | 0.82 (0.034) | 0.91 (0.046) | 0.69 (0.064) |
| | PC | 1.00 (0.005) | 0.76 (0.030) | 0.66 (0.104) | 0.41 (0.068) |
| | DAGMA | 0.99 (0.012) | 0.65 (0.047) | 0.91 (0.077) | 0.34 (0.066) |
| 4 | BOSS | 0.98 (0.012) | 0.82 (0.026) | 0.95 (0.031) | 0.75 (0.042) |
| | GRaSP | 0.99 (0.005) | 0.81 (0.025) | 0.95 (0.023) | 0.75 (0.041) |
| | fGES | 0.97 (0.019) | 0.80 (0.029) | 0.88 (0.037) | 0.71 (0.047) |
| | PC | 1.00 (0.000) | 0.70 (0.031) | 0.59 (0.051) | 0.37 (0.039) |
| | DAGMA | 0.99 (0.007) | 0.61 (0.051) | 0.87 (0.034) | 0.45 (0.067) |
| 6 | BOSS | 0.98 (0.013) | 0.81 (0.023) | 0.96 (0.027) | 0.77 (0.036) |
| | GRaSP | 0.99 (0.009) | 0.81 (0.021) | 0.97 (0.019) | 0.77 (0.032) |
| | fGES | 0.95 (0.018) | 0.74 (0.024) | 0.85 (0.037) | 0.65 (0.037) |
| | PC | 1.00 (0.003) | 0.64 (0.024) | 0.53 (0.043) | 0.32 (0.035) |
| | DAGMA | 1.00 (0.005) | 0.57 (0.042) | 0.89 (0.026) | 0.46 (0.048) |
| 8 | BOSS | 0.98 (0.010) | 0.80 (0.022) | 0.96 (0.017) | 0.77 (0.029) |
| | GRaSP | 0.99 (0.008) | 0.80 (0.019) | 0.97 (0.014) | 0.77 (0.025) |
| | fGES | 0.94 (0.015) | 0.65 (0.017) | 0.82 (0.050) | 0.56 (0.038) |
| | PC | 1.00 (0.003) | 0.57 (0.017) | 0.52 (0.050) | 0.28 (0.027) |
| | DAGMA | 1.00 (0.003) | 0.49 (0.066) | 0.92 (0.025) | 0.42 (0.068) |
| 10 | BOSS | 0.99 (0.007) | 0.81 (0.018) | 0.97 (0.013) | 0.78 (0.019) |
| | GRaSP | 0.99 (0.007) | 0.80 (0.019) | 0.97 (0.012) | 0.78 (0.019) |
| | fGES | 0.93 (0.017) | 0.57 (0.013) | 0.80 (0.041) | 0.48 (0.029) |
| | PC | 1.00 (0.003) | 0.51 (0.015) | 0.50 (0.051) | 0.24 (0.025) |
| | DAGMA | 1.00 (0.003) | 0.41 (0.077) | 0.92 (0.024) | 0.35 (0.081) |
| 12 | BOSS | 0.99 (0.008) | 0.80 (0.015) | 0.97 (0.018) | 0.79 (0.020) |
| | GRaSP | 0.99 (0.006) | 0.80 (0.013) | 0.98 (0.008) | 0.79 (0.016) |
| | fGES | 0.93 (0.010) | 0.50 (0.007) | 0.76 (0.036) | 0.41 (0.019) |
| | PC | 0.99 (0.005) | 0.45 (0.016) | 0.50 (0.040) | 0.21 (0.019) |
| | DAGMA | 1.00 (0.002) | 0.35 (0.056) | 0.91 (0.016) | 0.30 (0.055) |
| 14 | BOSS | 0.99 (0.005) | 0.80 (0.016) | 0.98 (0.009) | 0.78 (0.018) |
| | GRaSP | 0.99 (0.006) | 0.80 (0.015) | 0.98 (0.012) | 0.78 (0.016) |
| | fGES | 0.92 (0.016) | 0.43 (0.008) | 0.74 (0.041) | 0.35 (0.020) |
| | PC | 0.99 (0.004) | 0.40 (0.014) | 0.51 (0.040) | 0.19 (0.017) |
| | DAGMA | 1.00 (0.002) | 0.27 (0.053) | 0.90 (0.024) | 0.23 (0.054) |
| 16 | BOSS | 0.99 (0.008) | 0.79 (0.015) | 0.98 (0.014) | 0.78 (0.019) |
| | GRaSP | 0.99 (0.012) | 0.79 (0.012) | 0.98 (0.016) | 0.78 (0.014) |
| | fGES | 0.91 (0.018) | 0.38 (0.007) | 0.71 (0.038) | 0.29 (0.017) |
| | PC | 0.98 (0.010) | 0.34 (0.013) | 0.49 (0.050) | 0.16 (0.018) |
| | DAGMA | 1.00 (0.003) | 0.23 (0.046) | 0.88 (0.029) | 0.19 (0.051) |
| 18 | BOSS | 0.98 (0.007) | 0.79 (0.017) | 0.97 (0.011) | 0.77 (0.020) |
| | GRaSP | 0.98 (0.012) | 0.78 (0.017) | 0.97 (0.019) | 0.77 (0.018) |
| | fGES | 0.90 (0.019) | 0.34 (0.007) | 0.67 (0.046) | 0.25 (0.017) |
| | PC | 0.97 (0.014) | 0.31 (0.012) | 0.49 (0.042) | 0.15 (0.014) |
| | DAGMA | 1.00 (0.004) | 0.20 (0.031) | 0.87 (0.031) | 0.16 (0.032) |
| 20 | BOSS | 0.98 (0.009) | 0.78 (0.017) | 0.97 (0.011) | 0.77 (0.019) |
| | GRaSP | 0.99 (0.011) | 0.77 (0.017) | 0.98 (0.018) | 0.76 (0.021) |
| | fGES | 0.89 (0.028) | 0.30 (0.010) | 0.63 (0.041) | 0.21 (0.014) |
| | PC | 0.97 (0.012) | 0.27 (0.010) | 0.47 (0.042) | 0.12 (0.012) |
| | DAGMA | 0.99 (0.005) | 0.16 (0.021) | 0.83 (0.040) | 0.12 (0.021) |

Table 2: Erdős-Rényi - Mean (Standard Deviation) - 20 Repetitions

| Avg Deg | Algorithm | ΔBIC | Edges | Seconds |
|---|---|---|---|---|
| 2 | BOSS | −29.27 ( 12.132) | 85.05 ( 3.886) | 0.29 ( 0.058) |
| | GRaSP | −26.82 ( 13.207) | 83.05 ( 3.426) | 0.64 ( 0.268) |
| | fGES | −29.07 ( 11.938) | 84.35 ( 3.703) | 0.08 ( 0.018) |
| | PC | 315.45 ( 125.720) | 76.30 ( 3.164) | 0.04 ( 0.016) |
| | DAMGA | 257.81 ( 59.894) | 66.25 ( 4.778) | 24.43 ( 8.240) |
| 4 | BOSS | −31.87 ( 20.829) | 166.80 ( 6.110) | 0.59 ( 0.099) |
| | GRaSP | −32.87 ( 16.543) | 163.55 ( 5.316) | 0.72 ( 0.121) |
| | fGES | 59.78 ( 70.153) | 166.10 ( 6.528) | 0.12 ( 0.023) |
| | PC | 1497.21 ( 324.147) | 140.75 ( 6.290) | 0.05 ( 0.012) |
| | DAGMA | 835.05 ( 297.781) | 123.55 (10.190) | 19.97 ( 6.245) |
| 6 | BOSS | −37.73 ( 22.143) | 247.85 ( 7.734) | 1.07 ( 0.118) |
| | GRaSP | −46.35 ( 27.533) | 244.75 ( 6.568) | 1.22 ( 0.168) |
| | fGES | 514.82 ( 211.383) | 234.85 ( 8.387) | 0.19 ( 0.027) |
| | PC | 3434.62 ( 486.871) | 193.25 ( 6.866) | 0.12 ( 0.025) |
| | DAGMA | 1593.69 ( 528.149) | 172.80 (12.948) | 20.58 ( 7.643) |
| 8 | BOSS | −52.29 ( 32.968) | 327.95 ( 8.636) | 1.90 ( 0.192) |
| | GRaSP | −67.02 ( 28.251) | 323.70 ( 7.020) | 2.41 ( 0.244) |
| | fGES | 1629.07 ( 333.928) | 278.55 ( 6.117) | 0.26 ( 0.041) |
| | PC | 5456.39 ( 564.159) | 229.35 ( 6.777) | 0.22 ( 0.036) |
| | DAGMA | 3277.97 (1226.426) | 197.85 (26.442) | 20.27 ( 5.811) |
| 10 | BOSS | −67.96 ( 42.305) | 408.55 ( 9.971) | 3.23 ( 0.464) |
| | GRaSP | −73.79 ( 40.493) | 405.55 ( 9.784) | 4.52 ( 0.606) |
| | fGES | 3351.28 ( 525.921) | 307.65 ( 6.643) | 0.31 ( 0.038) |
| | PC | 7970.91 ( 792.056) | 255.60 ( 7.358) | 0.32 ( 0.039) |
| | DAGMA | 5983.26 (2160.954) | 203.70 (38.715) | 34.41 (4.913) |
| 12 | BOSS | −76.27 ( 61.569) | 488.20 ( 9.507) | 4.99 ( 0.762) |
| | GRaSP | −114.59 ( 31.515) | 484.55 ( 7.964) | 7.07 ( 1.233) |
| | fGES | 5546.54 ( 523.233) | 323.30 ( 4.281) | 0.40 ( 0.056) |
| | PC | 10445.14 ( 945.973) | 269.55 ( 9.545) | 0.43 ( 0.056) |
| | DAGMA | 8624.24 (2164.721) | 211.65 (33.585) | 31.87 (7.555) |
| 14 | BOSS | −89.58 ( 53.599) | 566.10 (11.836) | 7.48 ( 0.794) |
| | GRaSP | −111.92 ( 49.533) | 561.95 (10.450) | 10.59 ( 1.778) |
| | fGES | 7855.75 ( 678.104) | 330.45 ( 3.706) | 0.42 ( 0.047) |
| | PC | 12864.55 ( 944.373) | 279.45 (10.195) | 0.53 ( 0.059) |
| | DAGMA | 12696.05 (2397.839) | 192.25 (37.428) | 22.14 (3.549) |
| 16 | BOSS | −139.30 ( 68.319) | 643.30 (10.193) | 10.37 ( 0.731) |
| | GRaSP | −122.77 ( 90.846) | 639.85 (12.554) | 15.06 ( 2.132) |
| | fGES | 10013.10 ( 701.414) | 336.95 ( 2.373) | 0.45 ( 0.069) |
| | PC | 15568.38 ( 822.299) | 279.30 ( 9.050) | 0.55 ( 0.064) |
| | DAGMA | 15709.67 (2347.704) | 188.10 (36.784) | 17.98 ( 5.559) |
| 18 | BOSS | −87.23 ( 61.003) | 719.55 (17.111) | 14.08 ( 1.389) |
| | GRaSP | −42.03 ( 116.929) | 717.40 (16.282) | 19.36 ( 2.893) |
| | fGES | 12601.28 ( 911.461) | 340.25 ( 1.803) | 0.50 ( 0.055) |
| | PC | 18390.06 (1218.255) | 282.40 ( 9.162) | 0.67 ( 0.071) |
| | DAGMA | 19095.28 (2239.396) | 183.75 (28.703) | 19.83 ( 5.130) |
| 20 | BOSS | −112.38 ( 90.839) | 790.05 (15.998) | 17.89 ( 2.214) |
| | GRaSP | −121.13 ( 97.982) | 786.15 (15.267) | 25.49 ( 3.621) |
| | fGES | 14349.31 ( 935.139) | 342.25 ( 1.803) | 0.53 ( 0.065) |
| | PC | 20333.60 (1284.439) | 279.65 ( 9.281) | 0.72 ( 0.046) |
| | DAGMA | 23071.47 (2227.819) | 156.25 (20.326) | 20.42 ( 4.417) |

Table 3: Scale-Free - Mean (Standard Deviation) - 20 Repetitions

| Avg Deg | Algorithm | Adj Pre | Adj Rec | Ori Pre | Ori Rec |
|---|---|---|---|---|---|
| 2 | BOSS | 0.98 (0.018) | 0.82 (0.042) | 0.94 (0.044) | 0.70 (0.077) |
| | GRaSP | 0.99 (0.014) | 0.82 (0.038) | 0.95 (0.038) | 0.70 (0.075) |
| | fGES | 0.96 (0.034) | 0.81 (0.044) | 0.90 (0.081) | 0.69 (0.089) |
| | PC | 1.00 (0.003) | 0.75 (0.039) | 0.61 (0.085) | 0.39 (0.076) |
| | DAGMA | 0.99 (0.010) | 0.67 (0.038) | 0.92 (0.042) | 0.38 (0.077) |
| 4 | BOSS | 0.99 (0.009) | 0.83 (0.015) | 0.97 (0.021) | 0.77 (0.036) |
| | GRaSP | 0.99 (0.007) | 0.82 (0.016) | 0.96 (0.025) | 0.77 (0.037) |
| | fGES | 0.86 (0.067) | 0.71 (0.047) | 0.78 (0.081) | 0.63 (0.051) |
| | PC | 1.00 (0.004) | 0.67 (0.030) | 0.52 (0.072) | 0.31 (0.048) |
| | DAGMA | 1.00 (0.009) | 0.63 (0.041) | 0.92 (0.037) | 0.48 (0.057) |
| 6 | BOSS | 0.99 (0.008) | 0.81 (0.026) | 0.97 (0.017) | 0.77 (0.033) |
| | GRaSP | 0.99 (0.006) | 0.80 (0.026) | 0.96 (0.020) | 0.76 (0.033) |
| | fGES | 0.80 (0.060) | 0.59 (0.044) | 0.70 (0.072) | 0.51 (0.050) |
| | PC | 1.00 (0.004) | 0.57 (0.025) | 0.50 (0.086) | 0.26 (0.049) |
| | DAGMA | 1.00 (0.005) | 0.57 (0.041) | 0.91 (0.026) | 0.48 (0.052) |
| 8 | BOSS | 0.99 (0.007) | 0.81 (0.022) | 0.98 (0.013) | 0.79 (0.026) |
| | GRaSP | 0.99 (0.007) | 0.80 (0.021) | 0.97 (0.018) | 0.78 (0.029) |
| | fGES | 0.72 (0.051) | 0.47 (0.031) | 0.60 (0.069) | 0.40 (0.043) |
| | PC | 0.99 (0.007) | 0.47 (0.023) | 0.49 (0.051) | 0.22 (0.030) |
| | DAGMA | 1.00 (0.005) | 0.48 (0.052) | 0.90 (0.029) | 0.40 (0.055) |
| 10 | BOSS | 0.99 (0.007) | 0.79 (0.019) | 0.98 (0.013) | 0.78 (0.020) |
| | GRaSP | 0.99 (0.008) | 0.79 (0.019) | 0.97 (0.016) | 0.77 (0.021) |
| | fGES | 0.70 (0.036) | 0.41 (0.019) | 0.56 (0.048) | 0.32 (0.029) |
| | PC | 0.98 (0.009) | 0.41 (0.017) | 0.50 (0.057) | 0.20 (0.027) |
| | DAGMA | 1.00 (0.002) | 0.41 (0.056) | 0.90 (0.028) | 0.35 (0.059) |
| 12 | BOSS | 0.99 (0.006) | 0.79 (0.021) | 0.98 (0.010) | 0.78 (0.024) |
| | GRaSP | 0.99 (0.005) | 0.79 (0.018) | 0.97 (0.013) | 0.77 (0.022) |
| | fGES | 0.70 (0.045) | 0.36 (0.024) | 0.54 (0.047) | 0.28 (0.023) |
| | PC | 0.96 (0.015) | 0.35 (0.018) | 0.46 (0.035) | 0.16 (0.017) |
| | DAGMA | 1.00 (0.003) | 0.36 (0.050) | 0.88 (0.020) | 0.31 (0.046) |
| 14 | BOSS | 0.99 (0.005) | 0.79 (0.016) | 0.98 (0.008) | 0.78 (0.019) |
| | GRaSP | 0.99 (0.006) | 0.78 (0.017) | 0.97 (0.011) | 0.77 (0.020) |
| | fGES | 0.67 (0.029) | 0.30 (0.014) | 0.49 (0.044) | 0.22 (0.020) |
| | PC | 0.94 (0.019) | 0.30 (0.012) | 0.46 (0.046) | 0.14 (0.017) |
| | DAGMA | 0.99 (0.006) | 0.31 (0.034) | 0.86 (0.036) | 0.25 (0.031) |
| 16 | BOSS | 0.99 (0.005) | 0.79 (0.012) | 0.98 (0.008) | 0.77 (0.012) |
| | GRaSP | 0.98 (0.009) | 0.78 (0.011) | 0.97 (0.017) | 0.76 (0.015) |
| | fGES | 0.67 (0.027) | 0.27 (0.011) | 0.49 (0.028) | 0.20 (0.011) |
| | PC | 0.92 (0.017) | 0.27 (0.012) | 0.44 (0.065) | 0.12 (0.021) |
| | DAGMA | 0.99 (0.006) | 0.24 (0.037) | 0.86 (0.026) | 0.19 (0.036) |
| 18 | BOSS | 0.98 (0.005) | 0.77 (0.014) | 0.97 (0.006) | 0.76 (0.015) |
| | GRaSP | 0.98 (0.009) | 0.77 (0.014) | 0.97 (0.013) | 0.75 (0.017) |
| | fGES | 0.66 (0.031) | 0.24 (0.012) | 0.46 (0.038) | 0.17 (0.015) |
| | PC | 0.88 (0.023) | 0.23 (0.014) | 0.46 (0.069) | 0.11 (0.019) |
| | DAGMA | 0.99 (0.007) | 0.21 (0.043) | 0.83 (0.032) | 0.16 (0.038) |
| 20 | BOSS | 0.98 (0.007) | 0.76 (0.017) | 0.97 (0.010) | 0.75 (0.018) |
| | GRaSP | 0.98 (0.008) | 0.75 (0.015) | 0.97 (0.013) | 0.74 (0.016) |
| | fGES | 0.67 (0.037) | 0.22 (0.012) | 0.45 (0.040) | 0.15 (0.014) |
| | PC | 0.87 (0.029) | 0.21 (0.014) | 0.43 (0.064) | 0.10 (0.016) |
| | DAGMA | 0.98 (0.010) | 0.18 (0.035) | 0.81 (0.028) | 0.14 (0.032) |

Table 4: Scale-Free - Mean (Standard Deviation) - 20 Repetitions

| Avg Deg | Algorithm | ΔBIC | Edges | Seconds |
|---|---|---|---|---|
| 2 | BOSS | −26.55 ( 11.404) | 84.00 ( 4.078) | 0.30 (0.073) |
| | GRaSP | −23.59 ( 13.066) | 82.60 ( 4.018) | 0.50 (0.211) |
| | fGES | −0.54 ( 43.742) | 83.90 ( 4.518) | 0.07 (0.022) |
| | PC | 470.87 ( 144.483) | 75.40 ( 3.952) | 0.04 (0.050) |
| | DAGMA | 253.53 ( 47.175) | 67.90 ( 3.932) | 29.06 (9.857) |
| 4 | BOSS | −43.95 ( 15.748) | 166.85 ( 2.907) | 0.59 (0.088) |
| | GRaSP | −38.39 ( 18.982) | 165.50 ( 3.395) | 0.87 (0.167) |
| | fGES | 488.50 ( 242.279) | 163.80 ( 5.634) | 0.11 (0.020) |
| | PC | 2082.52 ( 410.197) | 133.55 ( 6.013) | 0.38 (0.348) |
| | DAGMA | 681.29 ( 213.554) | 126.85 ( 8.400) | 26.06 (8.459) |
| 6 | BOSS | −72.68 ( 16.589) | 245.20 ( 8.526) | 1.22 (0.210) |
| | GRaSP | −68.88 ( 24.561) | 243.35 ( 8.002) | 1.78 (0.418) |
| | fGES | 1643.97 ( 478.571) | 221.30 ( 6.997) | 0.17 (0.033) |
| | PC | 4396.89 ( 778.924) | 170.80 ( 7.090) | 0.54 (0.350) |
| | DAGMA | 1509.14 ( 651.626) | 172.95 (12.500) | 18.52 (4.705) |
| 8 | BOSS | −89.01 ( 27.280) | 327.95 ( 8.114) | 2.22 (0.292) |
| | GRaSP | −76.04 ( 31.893) | 325.50 ( 8.217) | 3.48 (0.452) |
| | fGES | 3686.06 ( 482.962) | 264.15 ( 7.073) | 0.26 (0.045) |
| | PC | 7147.10 ( 657.993) | 190.05 ( 8.953) | 1.74 (2.189) |
| | DAGMA | 3471.60 (1094.605) | 191.45 (20.997) | 20.76 (5.768) |
| 10 | BOSS | −112.15 ( 25.346) | 401.45 ( 8.906) | 3.65 (0.624) |
| | GRaSP | −97.90 ( 29.117) | 398.75 ( 8.271) | 6.32 (1.500) |
| | fGES | 5852.32 ( 740.949) | 289.85 ( 7.856) | 0.35 (0.073) |
| | PC | 9700.52 (1032.769) | 209.85 ( 8.216) | 1.59 (0.962) |
| | DAGMA | 5733.08 (1669.918) | 205.30 (27.891) | 21.77 (4.643) |
| 12 | BOSS | −133.19 ( 41.480) | 481.30 (11.282) | 5.34 (0.550) |
| | GRaSP | −112.36 ( 36.297) | 479.60 (10.002) | 11.10 (3.024) |
| | fGES | 7773.11 ( 700.188) | 306.45 ( 6.219) | 0.40 (0.058) |
| | PC | 12356.57 ( 757.713) | 220.85 ( 9.275) | 1.34 (1.007) |
| | DAGMA | 7951.55 (1554.948) | 218.40 (30.797) | 23.48 (8.366) |
| 14 | BOSS | −153.80 ( 37.574) | 558.30 (11.770) | 7.87 (0.954) |
| | GRaSP | −133.53 ( 38.246) | 555.15 (12.175) | 14.85 (2.982) |
| | fGES | 10081.52 ( 734.713) | 316.35 ( 5.678) | 0.47 (0.056) |
| | PC | 15099.54 ( 983.888) | 225.05 ( 7.244) | 1.16 (0.578) |
| | DAGMA | 10753.51 (1787.888) | 217.10 (24.531) | 35.74 (0.586) |
| 16 | BOSS | −158.52 ( 33.706) | 637.15 ( 9.686) | 11.14 (1.657) |
| | GRaSP | −115.00 ( 52.270) | 633.15 ( 9.213) | 22.15 (3.366) |
| | fGES | 12126.92 ( 687.750) | 325.35 ( 3.233) | 0.51 (0.074) |
| | PC | 17749.99 (1074.091) | 233.45 ( 8.593) | 0.92 (0.142) |
| | DAGMA | 15388.97 (2237.310) | 191.25 (29.958) | 28.44 (8.964) |
| 18 | BOSS | −199.97 ( 33.289) | 706.10 (13.286) | 14.15 (1.858) |
| | GRaSP | −157.95 ( 44.206) | 702.20 (12.968) | 28.48 (5.480) |
| | fGES | 13650.12 ( 878.170) | 328.10 ( 5.647) | 0.60 (0.073) |
| | PC | 19560.16 (1165.357) | 231.35 ( 9.115) | 0.92 (0.155) |
| | DAGMA | 17594.42 (2767.296) | 189.85 (39.175) | 30.24 (2.855) |
| 20 | BOSS | −232.22 ( 43.870) | 771.90 (14.418) | 17.12 (1.910) |
| | GRaSP | −191.62 ( 47.754) | 767.60 (13.228) | 33.99 (9.456) |
| | fGES | 15010.88 (1165.819) | 332.40 ( 3.817) | 0.57 (0.076) |
| | PC | 21362.04 (1513.351) | 238.05 (10.630) | 0.95 (0.191) |
| | DAGMA | 19822.75 (3327.041) | 186.15 (35.397) | 23.74 (5.430) |

Table 5: Gumbel - Mean (Standard Deviation) - 20 Repetitions

| Avg Deg | Algorithm | Adj Pre | Adj Rec | Ori Pre | Ori Rec |
|---|---|---|---|---|---|
| 2 | BOSS | 0.98 (0.021) | 0.77 (0.048) | 0.91 (0.040) | 0.58 (0.084) |
|  | DAGMA | 1.00 (0.004) | 0.58 (0.047) | 0.82 (0.061) | 0.47 (0.058) |
|  | LiNGAM | 0.86 (0.040) | 0.80 (0.045) | 0.80 (0.059) | 0.78 (0.050) |
| 4 | BOSS | 0.99 (0.008) | 0.75 (0.029) | 0.93 (0.029) | 0.67 (0.039) |
|  | DAGMA | 1.00 (0.002) | 0.54 (0.026) | 0.87 (0.026) | 0.46 (0.030) |
|  | LiNGAM | 0.89 (0.027) | 0.80 (0.029) | 0.86 (0.029) | 0.78 (0.031) |
| 6 | BOSS | 0.99 (0.008) | 0.76 (0.025) | 0.95 (0.019) | 0.71 (0.032) |
|  | DAGMA | 1.00 (0.000) | 0.50 (0.050) | 0.89 (0.026) | 0.45 (0.046) |
|  | LiNGAM | 0.90 (0.023) | 0.81 (0.024) | 0.88 (0.030) | 0.80 (0.026) |
| 8 | BOSS | 0.99 (0.007) | 0.77 (0.019) | 0.96 (0.013) | 0.73 (0.026) |
|  | DAGMA | 1.00 (0.002) | 0.46 (0.050) | 0.89 (0.027) | 0.41 (0.048) |
|  | LiNGAM | 0.91 (0.022) | 0.81 (0.017) | 0.89 (0.026) | 0.80 (0.020) |
| 10 | BOSS | 0.99 (0.008) | 0.76 (0.021) | 0.96 (0.013) | 0.73 (0.021) |
|  | DAGMA | 1.00 (0.001) | 0.39 (0.052) | 0.90 (0.018) | 0.35 (0.047) |
|  | LiNGAM | 0.90 (0.021) | 0.81 (0.023) | 0.89 (0.023) | 0.80 (0.024) |
| 12 | BOSS | 0.99 (0.006) | 0.75 (0.017) | 0.96 (0.012) | 0.72 (0.023) |
|  | DAGMA | 1.00 (0.001) | 0.32 (0.049) | 0.89 (0.016) | 0.29 (0.043) |
|  | LiNGAM | 0.90 (0.014) | 0.81 (0.015) | 0.88 (0.019) | 0.80 (0.017) |
| 14 | BOSS | 0.99 (0.005) | 0.74 (0.019) | 0.97 (0.011) | 0.72 (0.020) |
|  | DAGMA | 1.00 (0.004) | 0.28 (0.032) | 0.87 (0.032) | 0.25 (0.029) |
|  | LiNGAM | 0.90 (0.013) | 0.81 (0.018) | 0.89 (0.012) | 0.80 (0.019) |
| 16 | BOSS | 0.99 (0.005) | 0.72 (0.017) | 0.97 (0.013) | 0.70 (0.021) |
|  | DAGMA | 1.00 (0.003) | 0.24 (0.038) | 0.86 (0.019) | 0.20 (0.034) |
|  | LiNGAM | 0.90 (0.015) | 0.80 (0.014) | 0.89 (0.017) | 0.79 (0.015) |
| 18 | BOSS | 0.99 (0.004) | 0.73 (0.016) | 0.97 (0.010) | 0.71 (0.019) |
|  | DAGMA | 1.00 (0.004) | 0.21 (0.032) | 0.85 (0.028) | 0.18 (0.031) |
|  | LiNGAM | 0.90 (0.014) | 0.81 (0.015) | 0.89 (0.016) | 0.80 (0.016) |
| 20 | BOSS | 0.99 (0.004) | 0.72 (0.020) | 0.97 (0.009) | 0.70 (0.023) |
|  | DAGMA | 0.99 (0.006) | 0.20 (0.035) | 0.83 (0.033) | 0.17 (0.030) |
|  | LiNGAM | 0.91 (0.010) | 0.80 (0.014) | 0.90 (0.011) | 0.80 (0.015) |

Table 6: Gumbel - Mean (Standard Deviation) - 20 Repetitions

| Avg Deg | Algorithm | ΔBIC | Edges | Seconds |
|---|---|---|---|---|
| 2 | BOSS | −40.48 ( 14.664) | 78.55 ( 5.772) | 0.30 ( 0.058) |
| | DAGMA | 204.95 ( 76.866) | 58.20 ( 4.786) | 15.55 ( 6.018) |
| | LiNGAM | −54.83 ( 13.206) | 93.35 ( 7.415) | 105.82 ( 1.532) |
| 4 | BOSS | −65.33 ( 16.983) | 152.70 ( 5.741) | 0.51 ( 0.073) |
| | DAGMA | 565.94 ( 126.533) | 107.15 ( 5.153) | 13.78 ( 3.069) |
| | LiNGAM | −88.19 ( 18.882) | 179.85 ( 8.412) | 104.98 ( 1.320) |
| 6 | BOSS | −90.41 ( 24.002) | 231.80 ( 7.388) | 0.97 ( 0.141) |
| | DAGMA | 1229.57 ( 459.846) | 151.35 (15.034) | 16.07 ( 3.621) |
| | LiNGAM | −103.33 ( 28.705) | 271.40 (12.588) | 105.10 ( 0.483) |
| 8 | BOSS | −120.74 ( 24.131) | 310.15 ( 7.659) | 1.64 ( 0.167) |
| | DAGMA | 2291.71 ( 631.910) | 182.55 (19.888) | 16.97 ( 5.174) |
| | LiNGAM | −128.92 ( 26.895) | 357.05 (11.381) | 105.24 ( 0.611) |
| 10 | BOSS | −139.30 ( 35.203) | 382.95 (11.237) | 2.69 ( 0.443) |
| | DAGMA | 3743.50 (1029.405) | 194.05 (26.205) | 15.91 ( 5.753) |
| | LiNGAM | −169.05 ( 42.471) | 450.35 (17.187) | 105.66 ( 0.917) |
| 12 | BOSS | −166.78 ( 25.581) | 453.00 (10.110) | 4.01 ( 0.667) |
| | DAGMA | 5723.14 (1473.488) | 194.80 (29.269) | 17.11 ( 3.534) |
| | LiNGAM | −197.03 ( 35.866) | 542.25 (15.297) | 106.29 ( 1.778) |
| 14 | BOSS | −196.92 ( 34.669) | 521.00 (13.792) | 5.50 ( 0.634) |
| | DAGMA | 7252.63 (1027.255) | 199.95 (22.795) | 16.60 ( 3.512) |
| | LiNGAM | −236.48 ( 36.626) | 625.10 (16.905) | 110.73 ( 2.599) |
| 16 | BOSS | −238.72 ( 30.030) | 585.60 (13.141) | 6.93 ( 0.909) |
| | DAGMA | 9404.55 (1693.592) | 190.10 (30.805) | 18.25 ( 5.813) |
| | LiNGAM | −271.78 ( 37.077) | 711.60 (16.643) | 107.66 ( 3.950) |
| 18 | BOSS | −248.95 ( 44.181) | 664.15 (14.496) | 9.86 ( 1.305) |
| | DAGMA | 11752.12 (1845.939) | 191.40 (28.914) | 37.50 (31.535) |
| | LiNGAM | −285.60 ( 49.074) | 806.15 (21.475) | 105.50 ( 6.056) |
| 20 | BOSS | −278.55 ( 50.250) | 727.00 (19.960) | 12.69 ( 1.851) |
| | DAGMA | 13095.63 (2319.085) | 199.75 (35.909) | 22.59 (10.840) |
| | LiNGAM | −334.91 ( 44.193) | 887.20 (20.281) | 103.66 ( 3.493) |

Table 7: Exponential - Mean (Standard Deviation) - 20 Repetitions

| Avg Deg | Algorithm | Adj Pre | Adj Rec | Ori Pre | Ori Rec |
|---------|-----------|---------|---------|---------|---------|
| 2 | BOSS | 0.97 (0.016) | 0.84 (0.036) | 0.94 (0.030) | 0.71 (0.075) |
| | DAGMA | 0.99 (0.010) | 0.68 (0.050) | 0.85 (0.040) | 0.58 (0.051) |
| | LiNGAM | 0.88 (0.033) | 0.87 (0.034) | 0.84 (0.041) | 0.87 (0.039) |
| 4 | BOSS | 0.98 (0.014) | 0.81 (0.027) | 0.95 (0.018) | 0.75 (0.038) |
| | DAGMA | 1.00 (0.005) | 0.62 (0.033) | 0.88 (0.026) | 0.55 (0.036) |
| | LiNGAM | 0.89 (0.030) | 0.84 (0.023) | 0.88 (0.031) | 0.85 (0.023) |
| 6 | BOSS | 0.99 (0.008) | 0.80 (0.020) | 0.97 (0.014) | 0.77 (0.024) |
| | DAGMA | 1.00 (0.004) | 0.56 (0.046) | 0.90 (0.029) | 0.51 (0.046) |
| | LiNGAM | 0.91 (0.020) | 0.84 (0.020) | 0.90 (0.021) | 0.84 (0.019) |
| 8 | BOSS | 0.98 (0.010) | 0.81 (0.021) | 0.97 (0.015) | 0.79 (0.025) |
| | DAGMA | 1.00 (0.004) | 0.49 (0.047) | 0.90 (0.024) | 0.44 (0.042) |
| | LiNGAM | 0.91 (0.021) | 0.85 (0.018) | 0.90 (0.021) | 0.85 (0.019) |
| 10 | BOSS | 0.99 (0.007) | 0.80 (0.014) | 0.97 (0.013) | 0.78 (0.018) |
| | DAGMA | 1.00 (0.003) | 0.42 (0.055) | 0.90 (0.027) | 0.38 (0.051) |
| | LiNGAM | 0.91 (0.017) | 0.85 (0.015) | 0.91 (0.016) | 0.84 (0.015) |
| 12 | BOSS | 0.99 (0.006) | 0.79 (0.017) | 0.98 (0.006) | 0.77 (0.016) |
| | DAGMA | 1.00 (0.003) | 0.36 (0.046) | 0.89 (0.022) | 0.32 (0.041) |
| | LiNGAM | 0.91 (0.016) | 0.84 (0.015) | 0.91 (0.016) | 0.84 (0.014) |
| 14 | BOSS | 0.98 (0.006) | 0.78 (0.020) | 0.98 (0.009) | 0.77 (0.021) |
| | DAGMA | 0.99 (0.007) | 0.28 (0.036) | 0.86 (0.030) | 0.24 (0.031) |
| | LiNGAM | 0.90 (0.017) | 0.84 (0.015) | 0.90 (0.017) | 0.84 (0.015) |
| 16 | BOSS | 0.98 (0.007) | 0.77 (0.020) | 0.97 (0.011) | 0.76 (0.021) |
| | DAGMA | 0.99 (0.006) | 0.25 (0.041) | 0.84 (0.029) | 0.21 (0.039) |
| | LiNGAM | 0.90 (0.012) | 0.84 (0.011) | 0.90 (0.012) | 0.83 (0.012) |
| 18 | BOSS | 0.98 (0.007) | 0.77 (0.012) | 0.97 (0.010) | 0.76 (0.015) |
| | DAGMA | 0.99 (0.010) | 0.20 (0.037) | 0.83 (0.036) | 0.17 (0.031) |
| | LiNGAM | 0.90 (0.012) | 0.84 (0.007) | 0.90 (0.013) | 0.83 (0.007) |
| 20 | BOSS | 0.98 (0.004) | 0.76 (0.013) | 0.97 (0.006) | 0.75 (0.014) |
| | DAGMA | 0.98 (0.012) | 0.17 (0.043) | 0.80 (0.036) | 0.14 (0.035) |
| | LiNGAM | 0.91 (0.010) | 0.83 (0.009) | 0.90 (0.010) | 0.83 (0.009) |

Table 8: Exponential - Mean (Standard Deviation) - 20 Repetitions

| Avg Deg | Algorithm | $\Delta$BIC | Edges | Seconds |
|---|---|---|---|---|
| 2 | BOSS | −30.13 ( 13.273) | 86.40 ( 3.831) | 0.34 ( 0.058) |
| | DAGMA | 242.69 ( 83.387) | 68.85 ( 4.902) | 17.39 ( 4.199) |
| | LiNGAM | −44.29 ( 11.441) | 98.65 ( 5.234) | 103.83 ( 0.493) |
| 4 | BOSS | −53.80 ( 17.446) | 165.40 ( 5.041) | 0.62 ( 0.122) |
| | DAGMA | 671.26 ( 151.117) | 124.85 ( 6.854) | 15.10 ( 4.342) |
| | LiNGAM | −81.25 ( 19.414) | 189.25 ( 6.455) | 105.02 ( 0.398) |
| 6 | BOSS | −74.63 ( 19.124) | 245.00 ( 6.728) | 1.28 ( 0.165) |
| | DAGMA | 1554.85 ( 463.944) | 169.05 (13.904) | 15.83 ( 3.214) |
| | LiNGAM | −103.35 ( 13.446) | 277.60 ( 9.029) | 105.03 ( 0.618) |
| 8 | BOSS | −93.93 ( 26.702) | 329.55 ( 8.556) | 2.19 ( 0.258) |
| | DAGMA | 3077.68 ( 928.322) | 198.10 (19.123) | 17.80 ( 3.796) |
| | LiNGAM | −121.79 ( 25.327) | 374.35 (11.463) | 105.08 ( 0.397) |
| 10 | BOSS | −126.28 ( 26.061) | 405.45 ( 7.287) | 3.88 ( 0.483) |
| | DAGMA | 5243.46 (1178.323) | 211.15 (27.910) | 17.47 ( 5.457) |
| | LiNGAM | −153.33 ( 25.696) | 462.55 (11.569) | 105.54 ( 1.044) |
| 12 | BOSS | −140.74 ( 31.044) | 479.25 ( 9.808) | 5.78 ( 0.864) |
| | DAGMA | 7766.05 (1751.380) | 219.50 (28.065) | 18.54 ( 5.163) |
| | LiNGAM | −172.74 ( 27.491) | 552.15 ( 9.724) | 107.57 ( 1.642) |
| 14 | BOSS | −167.21 ( 33.004) | 557.30 (13.495) | 8.53 ( 1.297) |
| | DAGMA | 11865.65 (2010.847) | 197.95 (24.373) | 22.40 ( 8.998) |
| | LiNGAM | −203.28 ( 39.098) | 651.90 (16.814) | 109.91 ( 4.678) |
| 16 | BOSS | −167.46 ( 38.958) | 628.70 (14.120) | 11.39 ( 1.579) |
| | DAGMA | 14316.68 (2391.848) | 202.60 (33.271) | 19.48 ( 6.907) |
| | LiNGAM | −212.26 ( 30.366) | 739.25 (11.543) | 109.25 ( 3.127) |
| 18 | BOSS | −206.37 ( 30.259) | 702.35 (10.194) | 15.31 ( 1.998) |
| | DAGMA | 17855.62 (2605.919) | 184.10 (33.158) | 47.89 (30.300) |
| | LiNGAM | −249.31 ( 28.421) | 833.00 (15.731) | 111.06 ( 5.913) |
| 20 | BOSS | −221.56 ( 43.075) | 770.45 (13.221) | 19.62 ( 2.918) |
| | DAGMA | 21255.44 (3529.781) | 171.10 (43.489) | 23.55 ( 5.630) |
| | LiNGAM | −278.01 ( 31.714) | 921.60 (11.523) | 104.27 ( 0.655) |

Table 9: High Dimensional - Mean (Standard Deviation) - 10 Repetitions

| Variables | Algorithm | BIC $\lambda$ | Adj Pre | Adj Rec | Ori Pre | Ori Rec |
|---|---|---|---|---|---|---|
| 100 | BOSS | 2 | 0.98 (0.005) | 0.75 (0.008) | 0.97 (0.008) | 0.74 (0.009) |
| | GRaSP | 2 | 0.98 (0.004) | 0.74 (0.009) | 0.97 (0.010) | 0.73 (0.008) |
| | BOSS | 4 | 0.99 (0.003) | 0.64 (0.014) | 0.97 (0.008) | 0.62 (0.014) |
| | GRaSP | 4 | 0.99 (0.006) | 0.63 (0.016) | 0.96 (0.013) | 0.61 (0.018) |
| | BOSS | 8 | 0.98 (0.004) | 0.47 (0.009) | 0.94 (0.014) | 0.44 (0.014) |
| | GRaSP | 8 | 0.97 (0.007) | 0.45 (0.014) | 0.90 (0.023) | 0.41 (0.020) |
| 200 | BOSS | 2 | 0.98 (0.004) | 0.78 (0.013) | 0.98 (0.005) | 0.77 (0.013) |
| | GRaSP | 2 | 0.98 (0.004) | 0.78 (0.013) | 0.98 (0.005) | 0.77 (0.012) |
| | BOSS | 4 | 0.99 (0.002) | 0.69 (0.015) | 0.99 (0.005) | 0.68 (0.015) |
| | GRaSP | 4 | 0.99 (0.003) | 0.68 (0.015) | 0.98 (0.006) | 0.67 (0.018) |
| | BOSS | 8 | 0.99 (0.004) | 0.53 (0.015) | 0.97 (0.004) | 0.52 (0.016) |
| | GRaSP | 8 | 0.99 (0.004) | 0.52 (0.017) | 0.96 (0.007) | 0.51 (0.020) |
| 300 | BOSS | 2 | 0.98 (0.003) | 0.79 (0.006) | 0.98 (0.003) | 0.79 (0.007) |
| | GRaSP | 2 | 0.99 (0.003) | 0.79 (0.007) | 0.98 (0.004) | 0.79 (0.008) |
| | BOSS | 4 | 0.99 (0.002) | 0.71 (0.009) | 0.99 (0.002) | 0.70 (0.009) |
| | GRaSP | 4 | 1.00 (0.002) | 0.71 (0.009) | 0.99 (0.002) | 0.70 (0.009) |
| | BOSS | 8 | 0.99 (0.003) | 0.55 (0.011) | 0.98 (0.005) | 0.54 (0.012) |
| | GRaSP | 8 | 0.99 (0.003) | 0.55 (0.012) | 0.98 (0.007) | 0.54 (0.015) |
| 400 | BOSS | 2 | 0.98 (0.003) | 0.79 (0.005) | 0.97 (0.003) | 0.79 (0.005) |
| | GRaSP | 2 | 0.98 (0.003) | 0.79 (0.004) | 0.98 (0.003) | 0.79 (0.004) |
| | BOSS | 4 | 1.00 (0.002) | 0.71 (0.008) | 0.99 (0.003) | 0.70 (0.009) |
| | GRaSP | 4 | 1.00 (0.001) | 0.71 (0.008) | 0.99 (0.003) | 0.70 (0.009) |
| | BOSS | 8 | 0.99 (0.002) | 0.56 (0.012) | 0.99 (0.003) | 0.55 (0.014) |
| | GRaSP | 8 | 0.99 (0.002) | 0.55 (0.015) | 0.98 (0.004) | 0.54 (0.016) |
| 500 | BOSS | 2 | 0.98 (0.003) | 0.80 (0.005) | 0.97 (0.003) | 0.80 (0.005) |
| | GRaSP | 2 | 0.98 (0.003) | 0.80 (0.005) | 0.98 (0.003) | 0.80 (0.005) |
| | BOSS | 4 | 1.00 (0.001) | 0.72 (0.007) | 0.99 (0.002) | 0.72 (0.007) |
| | GRaSP | 4 | 1.00 (0.001) | 0.72 (0.007) | 0.99 (0.002) | 0.72 (0.008) |
| | BOSS | 8 | 1.00 (0.002) | 0.57 (0.008) | 0.99 (0.003) | 0.57 (0.008) |
| | GRaSP | 8 | 1.00 (0.002) | 0.57 (0.009) | 0.99 (0.004) | 0.56 (0.009) |
| 600 | BOSS | 2 | 0.98 (0.002) | 0.80 (0.005) | 0.97 (0.002) | 0.80 (0.005) |
| | GRaSP | 2 | 0.98 (0.002) | 0.80 (0.005) | 0.98 (0.002) | 0.80 (0.005) |
| | BOSS | 4 | 1.00 (0.001) | 0.72 (0.004) | 0.99 (0.001) | 0.72 (0.004) |
| | GRaSP | 4 | 1.00 (0.001) | 0.72 (0.005) | 0.99 (0.001) | 0.71 (0.005) |
| | BOSS | 8 | 1.00 (0.001) | 0.57 (0.009) | 0.99 (0.002) | 0.57 (0.009) |
| | GRaSP | 8 | 1.00 (0.001) | 0.57 (0.009) | 0.99 (0.002) | 0.56 (0.008) |

Table 10: High Dimensional - Mean (Standard Deviation) - 10 Repetitions

| Variables | Algorithm | BIC $\lambda$ | Adj Pre | Adj Rec | Ori Pre | Ori Rec |
|---|---|---|---|---|---|---|
| 700 | BOSS | 2 | 0.97 (0.003) | 0.80 (0.006) | 0.97 (0.003) | 0.80 (0.006) |
| | GRaSP | | 0.98 (0.002) | 0.80 (0.006) | 0.97 (0.003) | 0.80 (0.006) |
| | BOSS | 4 | 1.00 (0.001) | 0.72 (0.005) | 0.99 (0.001) | 0.72 (0.005) |
| | GRaSP | | 1.00 (0.001) | 0.72 (0.005) | 0.99 (0.001) | 0.72 (0.005) |
| | BOSS | 8 | 1.00 (0.001) | 0.58 (0.006) | 0.99 (0.002) | 0.58 (0.006) |
| | GRaSP | | 1.00 (0.001) | 0.58 (0.007) | 0.99 (0.002) | 0.57 (0.007) |
| 800 | BOSS | 2 | 0.97 (0.002) | 0.80 (0.005) | 0.97 (0.003) | 0.80 (0.005) |
| | GRaSP | | | | | |
| | BOSS | 4 | 1.00 (0.001) | 0.72 (0.005) | 0.99 (0.001) | 0.72 (0.005) |
| | GRaSP | | 1.00 (0.001) | 0.72 (0.006) | 0.99 (0.001) | 0.72 (0.005) |
| | BOSS | 8 | 1.00 (0.001) | 0.58 (0.005) | 0.99 (0.002) | 0.58 (0.005) |
| | GRaSP | | 1.00 (0.001) | 0.58 (0.006) | 0.99 (0.002) | 0.57 (0.006) |
| 900 | BOSS | 2 | 0.97 (0.002) | 0.80 (0.004) | 0.97 (0.002) | 0.80 (0.004) |
| | GRaSP | | | | | |
| | BOSS | 4 | 1.00 (0.001) | 0.73 (0.004) | 1.00 (0.001) | 0.72 (0.004) |
| | GRaSP | | 1.00 (0.000) | 0.73 (0.004) | 1.00 (0.001) | 0.72 (0.004) |
| | BOSS | 8 | 1.00 (0.001) | 0.59 (0.006) | 1.00 (0.001) | 0.58 (0.006) |
| | GRaSP | | 1.00 (0.001) | 0.58 (0.006) | 0.99 (0.002) | 0.58 (0.006) |
| 1000 | BOSS | 2 | 0.97 (0.002) | 0.80 (0.003) | 0.97 (0.002) | 0.8 (0.003) |
| | GRaSP | | | | | |
| | BOSS | 4 | 1.00 (0.001) | 0.73 (0.006) | 1.0 (0.002) | 0.72 (0.006) |
| | GRaSP | | 1.00 (0.001) | 0.72 (0.006) | 1.00 (0.001) | 0.72 (0.006) |
| | BOSS | 8 | 1.00 (0.001) | 0.59 (0.006) | 1.00 (0.001) | 0.58 (0.006) |
| | GRaSP | | 1.00 (0.001) | 0.59 (0.006) | 1.00 (0.001) | 0.58 (0.006) |

Table 11: High Dimensional - Mean (Standard Deviation) - 10 Repetitions

| Variables | Algorithm | BIC $\lambda$ | $\Delta$BIC | Edges | Seconds |
|---|---|---|---|---|---|
| 100 | BOSS | 2 | −250.40 (37.438) | 762.7 (6.533) | 16.88 (2.089) |
| | GRaSP | | −219.89 (38.486) | 756.4 (9.419) | 29.21 (3.756) |
| | BOSS | 4 | 680.01 (115.182) | 646.3 (14.568) | 5.16 (0.747) |
| | GRaSP | | 748.13 (121.240) | 642.3 (14.659) | 9.94 (1.854) |
| | BOSS | 8 | 4065.45 (294.262) | 477.6 (9.857) | 1.74 (0.255) |
| | GRaSP | | 4826.93 (546.131) | 458.4 (13.032) | 2.61 (0.434) |
| 200 | BOSS | 2 | −449.04 (50.265) | 1587.0 (20.827) | 110.62 (11.576) |
| | GRaSP | | −401.91 (81.173) | 1578.2 (22.115) | 226.31 (17.332) |
| | BOSS | 4 | 1062.78 (139.848) | 1389.9 (28.938) | 27.41 (3.499) |
| | GRaSP | | 1234.52 (171.907) | 1377.0 (29.405) | 59.14 (6.809) |
| | BOSS | 8 | 7386.30 (418.935) | 1069.0 (32.156) | 8.74 (0.559) |
| | GRaSP | | 7943.37 (357.622) | 1054.8 (35.333) | 16.02 (2.520) |
| 300 | BOSS | 2 | −697.19 (58.726) | 2428.8 (15.852) | 320.69 (24.340) |
| | GRaSP | | −650.07 (44.280) | 2417.1 (20.091) | 689.65 (77.067) |
| | BOSS | 4 | 1391.78 (159.564) | 2141.4 (24.613) | 71.44 (6.100) |
| | GRaSP | | 1487.97 (163.380) | 2133.7 (25.578) | 155.94 (15.237) |
| | BOSS | 8 | 10668.10 (640.734) | 1666.2 (32.519) | 21.79 (1.769) |
| | GRaSP | | 11103.41 (693.257) | 1651.4 (35.034) | 59.82 (10.100) |
| 400 | BOSS | 2 | −989.29 (116.766) | 3245.7 (15.384) | 584.74 (57.027) |
| | GRaSP | | −945.61 (9.502) | 3231.2 (17.869) | 382.63 (57.505) |
| | BOSS | 4 | 1946.53 (303.747) | 2843.5 (30.642) | 133.17 (9.047) |
| | GRaSP | | 2029.54 (257.483) | 2834.7 (31.489) | 330.64 (19.780) |
| | BOSS | 8 | 13366.98 (868.069) | 2245.6 (50.000) | 43.28 (3.978) |
| | GRaSP | | 14230.16 (1053.206) | 2218.4 (57.618) | 114.04 (11.378) |
| 500 | BOSS | 2 | −1282.97 (48.490) | 4102.1 (26.860) | 1012.22 (60.835) |
| | GRaSP | | −1239.31 (36.450) | 4085.2 (29.001) | 2331.55 (249.112) |
| | BOSS | 4 | 2168.07 (253.685) | 3619.0 (34.438) | 214.01 (9.017) |
| | GRaSP | | 2202.18 (277.513) | 3609.7 (36.794) | 637.89 (61.230) |
| | BOSS | 8 | 16143.34 (783.084) | 2880.1 (37.513) | 77.40 (7.527) |
| | GRaSP | | 16946.30 (1096.986) | 2856.2 (40.386) | 233.62 (23.181) |
| 600 | BOSS | 2 | −1573.63 (118.084) | 4924.2 (28.358) | 1483.88 (97.721) |
| | GRaSP | | −1542.92 (108.987) | 4909.1 (30.443) | 3755.78 (677.803) |
| | BOSS | 4 | 2599.43 (184.187) | 4332.9 (29.027) | 320.71 (22.809) |
| | GRaSP | | 2713.14 (166.374) | 4319.4 (28.776) | 1018.19 (101.612) |
| | BOSS | 8 | 18967.19 (779.458) | 3450.9 (53.978) | 117.72 (12.370) |
| | GRaSP | | 19657.44 (849.114) | 3426.7 (55.140) | 427.71 (42.478) |

Table 12: High Dimensional - Mean (Standard Deviation) - 10 Repetitions

| Variables | Algorithm | BIC $\lambda$ | $\Delta$BIC | Edges | Seconds |
|---|---|---|---|---|---|
| 700 | BOSS | 2 | −1899.57 (134.364) | 5773.7 (33.036) | 2453.32 (100.902) |
| | GRaSP | | −1865.52 (120.199) | 5749.2 (37.611) | 5758.49 (564.612) |
| | BOSS | 4 | 2995.26 (240.846) | 5067.6 (39.973) | 456.97 (28.944) |
| | GRaSP | | 3023.06 (225.375) | 5056.4 (38.208) | 1430.72 (106.472) |
| | BOSS | 8 | 21327.81 (719.142) | 4086.1 (43.144) | 178.79 (11.971) |
| | GRaSP | | 21783.71 (890.414) | 4067.9 (44.851) | 698.20 (71.643) |
| 800 | BOSS | 2 | −2210.01 (87.533) | 6617.1 (37.072) | 3083.87 (225.958) |
| | GRaSP | | | | |
| | BOSS | 4 | 3395.12 (215.775) | 5795.3 (42.008) | 702.54 (55.730) |
| | GRaSP | | 3547.25 (256.39) | 5777.7 (46.159) | 2103.76 (167.556) |
| | BOSS | 8 | 24391.82 (886.115) | 4668.5 (40.768) | 282.38 (23.453) |
| | GRaSP | | 24820.67 (708.219) | 4650.9 (42.956) | 1052.97 (87.079 |
| 900 | BOSS | 2 | −2579.65 (113.671) | 7470.1 (32.306) | 4051.92 (113.881) |
| | GRaSP | | | | |
| | BOSS | 4 | 3615.29 (226.659) | 6556.4 (30.310) | 880.03 (47.222) |
| | GRaSP | | 3763.39 (257.163) | 6539.1 (31.125) | 2832.26 (196.926) |
| | BOSS | 8 | 27255.29 (865.222) | 5284.7 (50.352) | 471.74 (43.150) |
| | GRaSP | | 27741.60 ( 978.542) | 5262.7 (55.201) | 1508.55 (162.988) |
| 1000 | BOSS | 2 | −2921.0 (153.220) | 8315.8 (22.220) | 5113.12 (440.444) |
| | GRaSP | | | | |
| | BOSS | 4 | 4071.33 (371.126) | 7274.1 (54.519) | 1113.69 (61.362) |
| | GRaSP | | 4246.67 (452.957) | 7257.4 (57.57) | 3730.44 (375.983) |
| | BOSS | 8 | 29520.77 (1101.301) | 5889.5 (57.853) | 599.24 (43.804) |
| | GRaSP | | 30005.90 (1233.898) | 5870.1 (60.090) | 2176.70 (203.593) |

Table 13: fMRI Simulation - Mean (Standard Deviation) - 40 Repetitions

| Algorithm | Adj Pre | Adj Rec | Ori Pre | Ori Rec |
|---|---|---|---|---|
| BOSS | 0.99 (0.005) | 0.94 (0.009) | 0.96 (0.009) | 0.90 (0.012) |
| DAGMA | 1.00 (0.000) | 0.69 (0.038) | 0.98 (0.005) | 0.67 (0.037) |
| fGES | 0.97 (0.007) | 0.60 (0.012) | 0.70 (0.038) | 0.43 (0.025) |
| LiNGAM | 0.54 (0.026) | 0.94 (0.008) | 0.35 (0.031) | 0.62 (0.032) |

Table 14: fMRI Simulation - Mean (Standard Deviation) - 40 Repetitions

| Algorithm | $\Delta$BIC | Edges | Seconds |
|---|---|---|---|
| BOSS | 211.79 ( 85.026) | 951.40 (23.546) | 15.46 ( 1.987) |
| DAGMA | 3080.85 (853.227) | 687.15 (40.417) | 54.58 (18.957) |
| fGES | 7784.48 (751.306) | 617.35 ( 7.033) | 5.12 ( 1.344) |
| LiNGAM | 3868.93 (376.249) | 1752.75 (99.268) | 582.03 ( 3.278) |

Table 15: fMRI - Mean (Standard Deviation) - 171 Scans

| Algorithm | BIC | Undirected Edges | Total Edges |
|---|---|---|---|
| BOSS | 98752.23 (25139.907) | 3.09 (2.189) | 2644.61 (430.419) |
| fGES | 96171.56 (24592.247) | 5.79 (7.083) | 2452.94 (394.0745) |

Table 16: fMRI - Mean (Standard Deviation) - 171 Scans

| $\Delta$ Algorithm | $\Delta$ BIC | $\Delta$ Edges |
|---|---|---|
| BOSS − fGES | 2580.68 (664.574) | 191.67 (55.961) |