# OpenReview forum: "Fast Scalable and Accurate Discovery of DAGs Using the Best Order Score Search and Grow Shrink Trees"
_NeurIPS.cc/2023/Conference — NeurIPS 2023 poster_

### Official Review · Reviewer_oAsg · 2023-07-02

**Soundness:** 3 good
**Presentation:** 3 good
**Contribution:** 3 good
**Rating:** 4
**Confidence:** 2

**Summary:**

This paper introduces the best order score search (BOSS) algorithm and grow-shrink trees (GSTs) for learning directed acyclic graphs (DAGs) in machine learning and causal discovery. BOSS achieves state-of-the-art performance in accuracy and execution time, making it a valuable tool for problems with highly connected variables. The paper also applies BOSS to resting-state fMRI data, demonstrating its practicality and effectiveness. The algorithm is designed to be a more efficient alternative to the existing Greedy Sparsest Permutation (GRaSP) algorithm.

**Strengths:**

- The paper introduces the concept of Grow-Shrink Trees (GSTs), which are tree data structures for caching the results of the grow and shrink subroutines of GS. This data structure is compatible with many permutation-based structure learning algorithms, including BOSS and GRaSP. The use of GSTs is a novel approach that efficiently stores information needed for running GS.
- The paper is well-written and clear. It provides detailed explanations and examples, making it easy to understand the proposed algorithm and its components. The use of figures and tables also helps to illustrate the concepts and results.
- The pape compares BOSS against other algorithms such as GRaSP, fGES, PC, and DAGMA on linear Gaussian data generated from Erd˝os-Rényi and scale-free networks. The results show that BOSS maintains a high level of accuracy while scaling much better than GRaSP, indcating that it could be a valuable tool in the field of structure learning.

**Weaknesses:**

- One potential weakness is the lack of discussion on the limitations of the BOSS algorithm and GSTs. While the paper acknowledges that there is room for additional improvements, it does not provide a detailed analysis of the limitations of the proposed approach. For example, the paper could discuss the assumptions made by BOSS, such as causal sufficiency, and how these assumptions may affect the accuracy and scalability of the algorithm. Additionally, the paper could explore the applicability of BOSS to other types of data beyond fMRI, such as financial data or electronic health records. Addressing these limitations would provide a more comprehensive understanding of the strengths and weaknesses of the proposed approach.
- Validation on more Real-World Data: The validation of the algorithm on fMRI data is a good step, but more extensive validation on diverse real-world datasets could strengthen the robustness of the findings.

**Questions:**

- How does the BOSS algorithm handle the problem of confounding variables in fMRI data, and how does it ensure that the recovered causal networks are not biased by these variables?
- Would there be non-linear causal relationships in fMRI data, and how does it ensure that the recovered causal networks capture these relationships accurately?
- How does the performance of the BOSS algorithm vary with the level of sparsity in the underlying causal graph in fMRI data?
- Would the performance of the model vary with the level of noise in data?
- Discussion on Limitations: Could you include a section discussing the limitations of your work and potential areas for future research?
- Results Interpretation: Could you provide more insight into how you interpreted your results and why you drew the conclusions you did? This would help to understand the implications of your findings.



**Limitations:**

- The authors have discussed the limitations of their work in the context of the performance of the BOSS algorithm. They have compared it with other algorithms like GRaSP, fGES, PC, and DAGMA on linear Gaussian data and have also tested it on non-Gaussian data. They have acknowledged that while BOSS performs well in terms of BIC score and running time, its recall may be low in certain scenarios.
Question: You have mentioned that BOSS has a low recall in certain scenarios. Could you elaborate on these scenarios? How could this limitation be addressed in future work?
- It would be beneficial to include a section in your paper discussing the ethical implications of your work. This could include potential misuse of your algorithm, biases in the data that could affect the algorithm's performance, and the implications of incorrect predictions by the algorithm.

---

> ### Author Rebuttal · Authors · 2023-08-10
>
> Apologies for truncating a few of your comments, we ran out of characters for our rebuttal.
>
> **One potential weakness is...**
>
> Thanks for your comments and criticisms; we have found them helpful and constructive! See our responses to your points below.
>
> **Validation on more Real-World Data: The validation of the algorithm on fMRI data is a good step, but more extensive validation on diverse real-world datasets could strengthen the robustness of the findings.**
>
> We hope to apply this algorithm to many real-world datasets in the future; however, we lack the space to include multiple examples in the current paper. Regarding the choice of real-world application, fMRI was used due to the low recall of other methods in highly connected systems. The high recall of BOSS and GRaSP in these problems is not trivial, and GRaSP could not scale to the number of variables in most parcellations of fMRI data before this paper. We also think it is worth noting that the fMRI application included in the paper ran the BOSS algorithm on 171 datasets with 379 columns and 520 rows, so the included application was not a small undertaking.
>
>
> **How does the BOSS algorithm handle the problem of confounding variables in fMRI data, and how does it ensure that the recovered causal networks are not biased by these variables?**
>
> Thanks for the question; we refer you to our responses to the other reviewers on this point. In summary, we expect to find extra edges, but they can be removed using post-processing techniques which we leave for future work.
>
> **Would there be non-linear causal relationships in fMRI data, and how does it ensure that the recovered causal networks capture these relationships accurately?**
>
> In general, fMRI data look fairly linear; however, it should be noted that the linearity assumption is made for the Gaussian BIC score and not for the BOSS algorithm itself. The Gaussian BIC score can be swapped out for another consistent score that models non-linearities in the data, and correctness will hold. However, it should be noted that such scores are often much slower, so the algorithm will not scale as well in this case. Lastly, we note that the dominant technique used to analyze function connectivity in fMRI data is to threshold the sample correlation matrix, which also does not handle non-linearity.
>
> **How does the performance of the BOSS algorithm vary with the level of sparsity in the underlying causal graph in fMRI data?**
>
> It is hard to analyze how BOSS deals with varying levels of sparsity in fMRI data since the true underlying structure is unknown. That being said, we believe the results that look at varying levels of sparsity for the scale-free linear cases should give an idea. Indeed, the method by which we simulated the scale-free aspect of the graph was inspired by the fMRI data. Figures 3 and 4 show the performance of BOSS (and other algorithms) as a function of the average degree of the underlying model. For the range of parameters we considered, the performance of BOSS (precision and recall for adjacencies and orientations) was essentially unaffected by variation in sparsity.
>
>
> **Would the model's performance vary with the noise level in the data?**
>
> If the noise is increased in the exogenous noise terms, it does not appear to have a substantial effect; we’ve tested this for a wide range of these parameters (though there wasn’t room to include a formal result for this in the paper). If the noise is a large additional measurement noise, we have not tested this and plan to in future work. We have added this comment to the paper.  Small added Gaussian measurement noise has been unproblematic in our testing.
>
>
> **Discussion on Limitations: Could you include a section discussing the limitations of your work and potential areas for future research?**
>
> We have collected up the various comments regarding the limitations of the work and potential areas of future research and will include them in a new section.
>
>
> **Results Interpretation: Could you provide more insight into how you interpreted your results and why you drew the conclusions you did? This would help to understand the implications of your findings.**
>
> The results on the fMRI data is mainly included as a proof of concept. We hoped it would demonstrate that this algorithm can be applied to such data and the comments regarding the scale-free nature of the learned output are mainly to note that we are learning something that is consistent with the accepted nature of brain connectivity. We are currently working on several applied projects using this algorithm, where we plan to dive deeply into what we have discovered using this algorithm. This paper is intended to introduce the algorithm but not dive deeply into any data analysis; we leave that to future work.
>
> **The authors have discussed the limitations of their work in the context of...**
>
> The performance of BOSS drops off in Table 2 (Figure 4b) when the penalty discount (BIC lambda) is increased. This was done for the sake of comparison to the GRaSP algorithm, which required the lower penalty discount in order to return in a reasonable amount of time (under an hour). However, there is no reason to not also report the results of the BOSS algorithm run on these data with a lower penalty discount. We expect the recall of BOSS to improve by doing so and plan to add these results to Table 2 (Figure 4b). The performance of BOSS could also degrade when its assumptions are violated, and we will include a discussion on this in the added section in limitations.
>
>
> **It would be beneficial...**
>
> Thanks for the suggestion, we will comment on the ethical implications of misusing / misinterpreting the results of our algorithm as you suggest with / after the added discussion of limitations.

---

> ### Comment · Reviewer_oAsg · 2023-08-21
> **Response to Rebuttal**
>
> Thanks for the authors for the additonal explanation. I believe adding those details can further help readers better understand the work. I would like to remain my original ratings.

---

### Official Review · Reviewer_P8PW · 2023-07-03

**Soundness:** 2 fair
**Presentation:** 2 fair
**Contribution:** 2 fair
**Rating:** 5
**Confidence:** 1

**Summary:**

It's beyond my knowledge so please disregard my scores and reviews.

**Strengths:**

It's beyond my knowledge so please disregard my scores and reviews.

**Weaknesses:**

It's beyond my knowledge so please disregard my scores and reviews.

**Questions:**

It's beyond my knowledge so please disregard my scores and reviews.

**Limitations:**

It's beyond my knowledge so please disregard my scores and reviews.

---

> ### Author Rebuttal · Authors · 2023-08-10
>
> Thank you for your honesty!

---

### Official Review · Reviewer_yEGg · 2023-07-06

**Soundness:** 3 good
**Presentation:** 3 good
**Contribution:** 3 good
**Rating:** 6
**Confidence:** 2

**Summary:**

The authors introduce two novel methods for learning Directed Acyclic Graphs (ADGs): best order score search or BOSS and grow-shrink
trees or GSTs. These methods have a similar performance to state-of-the-art approaches (namely GRaSP and also others: fGES, DAGMA, LiNGAM, etc), but the authors proof that are less computationally intensive and therefore more scalable. The authors then validate their methods by learning brain networks from synthetic and real brain recordings (fMRI).

**Strengths:**

- originality: the paper introduces two novel approaches for learning DAGs. Even though this work closely follows previous studies on permutation-based structure learning, it also presents significant new departures from the previous algorithms.

- quality: the theoretical proofs seem solid and the methods are backed with a reasonable amount of numerical and experimental evidence.

- clarity: the paper is very concisely and clearly written.  Although a lot of the information is in the supplements.

- significance: learning network from brain recordings is a very useful tool to understanding the brain and its pathologies. A method that significantly increases scalability with same accuracy as the state-of-the-art is therefore noteworthy.

**Weaknesses:**

- Validation on real fMRI is (somewhat understandably) weak because the scale free connectivity of the brain remains a theory rather than a proven fact.

- Many details of the work are left out from the main paper and included in the supplemental material. It is almost hard to understand the main paper without reading the supplements.

**Questions:**

No questions.

**Limitations:**

Authors have adequately addressed the limitations of their work.

---

> ### Author Rebuttal · Authors · 2023-08-10
>
> **Validation on real fMRI is (somewhat understandably) weak because the scale-free connectivity of the brain remains a theory rather than a proven fact.**
>
> Thank you for your comment, we will add references (a few relevant references are listed below) to bolster this particular point. These references give more recent, strong positive support for this position.
>
> Lynn, C. W., & Bassett, D. S. (2019). The physics of brain network structure, function and control. Nature Reviews Physics, 1(5), 318-332.
>
> Galinsky, V. L., & Frank, L. R. (2017). A unified theory of neuro-MRI data shows scale-free nature of connectivity modes. Neural computation, 29(6), 1441-1467.
>
> Mansour L, S., Di Biase, M. A., Smith, R. E., Zalesky, A., & Seguin, C. (2023). Connectomes for 40,000 UK Biobank participants: A multi-modal, multi-scale brain network resource. bioRxiv, 2023-03.
>
> Hanson, S. J., Mastrovito, D., Hanson, C., Ramsey, J., & Glymour, C. (2016). Scale-free exponents of resting state provide a biomarker for typical and atypical brain activity. arXiv preprint arXiv:1605.09282.
>
> Zhang, A., Fang, J., Liang, F., Calhoun, V. D., & Wang, Y. P. (2018). Aberrant brain connectivity in schizophrenia detected via a fast gaussian graphical model. IEEE journal of biomedical and health informatics, 23(4), 1479-1489.
>
> Grosu, G. F., Hopp, A. V., Moca, V. V., Bârzan, H., Ciuparu, A., Ercsey-Ravasz, M., ... & Mureșan, R. C. (2023). The fractal brain: scale-invariance in structure and dynamics. Cerebral Cortex, 33(8), 4574-4605.
>
>
> **Many details of the work are left out of the main paper and included in the supplemental material. It is almost hard to understand the main paper without reading the supplements.**
>
> Thank you for this comment, the supplement contains details about: (1) how the data were simulated, (2) the parameters chosen for the various algorithms, and (3) tables giving the exact values of the results that are depicted as plots in the main paper. We included these details in the supplement for readers who wish to delve more into the details of our experiments, and we do not feel that the paper is incomplete without them. Could you be more precise about which aspect of the supplement you think should be included in the main paper?

---

### Official Review · Reviewer_9NHG · 2023-07-27

**Soundness:** 3 good
**Presentation:** 3 good
**Contribution:** 2 fair
**Rating:** 5
**Confidence:** 1

**Summary:**

The paper proposes a computationally efficient algorithm using the grow-shrink trees to iterate through some of the combinatoric number of directed acyclic graphs in a graphical model. The approach builds on early work but is faster.

**Strengths:**

The paper is straightforward, easy to read (examples are given), and focused in its presentation. The results show that it is faster than the baseline GRaSP.



**Weaknesses:**

The paper's focus on the approach also means there is not much perspective given until the final discussion.

The paper's claim of scaling to thousands is not thoroughly tested (results for 1000 appear to take 500 s). It is not clear what is necessary to apply the approach to higher-resolution parcellations.

**Questions:**

1. What is the computational complexity of the algorithm?
2.  Proposition 4 states that the initial permutation doesn't matter if some conditions are met. In practice, if these conditions are unmet or unknown, does the initial permutation matter?


**Limitations:**

Limitations are discussed.

---

> ### Author Rebuttal · Authors · 2023-08-10
>
> **The paper's focus on the approach also means little perspective is given until the final discussion.**
>
> We will add more details to / rework the introduction to help readers understand our perspective. In order to help us do this, could you describe what perspective you gained in the final discussion that was missing in the introduction?
>
> **The paper's claim of scaling to thousands is not thoroughly tested (results for 1000 appear to take 500 s). It is unclear what is necessary to apply the approach to higher-resolution parcellations.**
>
> Our intention was not to claim that our algorithm can scale to thousands of variables but rather to demonstrate that we can scale to at least 1000 variables on a laptop in under 10 minutes — this was done on a single processor and could be further sped up by using multiple processors in parallel. The paper presents an algorithm that can scale to densely connected problems an order of magnitude larger than what was previously possible in terms of variables (GRaSP without the use of GSTs does not scale much past 100 variables). As far as we know, fMRI parcellations in the literature rarely have more than 1000 variables.
>
> **What is the computational complexity of the algorithm?**
>
> We will add a discussion about the algorithm’s complexity and running time. In the worst case, the main loop will be performed no more than n times, where n is the number of variables. Although, in our experiments, BOSS usually only repeated the main loop two or three times. Each main loop requires 3n^2 runs of the grow shrink subroutine, so BOSS will make O(n^3) calls to the GSTs. While profiling our algorithm, we found that it spends nearly all of its execution time performing such GST calls. These calls are cached so redundant regressions that need to be performed while running similar grow shrink subroutines can simply be looked up rather than recalculated. Indeed, one can show that once a regression is performed, it will never need to be run again.
>
> **Proposition 4 states that the initial permutation doesn't matter if some conditions are met. In practice, does the initial permutation matter if these conditions are unmet or unknown?**
>
> In general, it is possible to run the entire algorithm multiple times from different initial permutations and take the result that yields the most optimal BIC score; our software has a parameter “number of runs” that allows the user to do this. In our simulations, when the ground truth is a DAG, there is little to no substantial advantage to consider multiple starting points. However, assuming that the ground truth is a DAG is admittedly a bad assumption for most real world datasets. In the causally insufficient case, regardless of the initial permutation, we can expect to learn a few extra edges. This can be corrected if the user post-processes the learned graph with the FCI algorithm, like what is done in the GFCI algorithm; we leave this investigation to future work.
>
> Also, relative to confounding and fMRI, together with the whole-brain coverage, I would add something about the preprocessing that was used for this real fMRI data, in particular, what is commonly referred to as nuisance regression, which implies regressing the bold signal out of movements, trends, white matter, and csf artifacts, etc. Both strategies strongly mitigate the effect of possible confounders.

---

> > ### Comment · Reviewer_9NHG · 2023-08-14
> > **I've read the rebuttal.**
> >
> > **The paper's focus on the approach also means little perspective is given until the final discussion.**
> > In the discussion the assumptions and limitations are stated.
> > In the discussion the family of graphs (ER and scale-free) underlying the DAG for the synthetic are mentioned.
> >
> > **The paper's claim of scaling**
> > Although parcellations with more parcel's aren't common, couldn't arbitrarily small parcels be created down to the individual voxels through various techniques? My point is that it should be clarified how scalable the proposed techniques are, even if real data is not readily available.
> >
> > **What is the computational complexity**
> > Thanks.
> >
> > **Proposition 4**
> > The option to run it multiple times and the motivation (not assuming a DAG) should be clearly stated in the method section, not just the software, as should be the inclusion of fast causal inference as post-processing.

---

> > > ### Comment · Reviewer_9NHG · 2023-08-14
> > > **confidence**
> > >
> > > I'll also add that my confidence is due to my own lack of in-depth familiarity with the causal discovery material.
> > > It is unfortunate that none of the reviewers are confident. I was added as an emergence reviewer and am not familiar with previous work: permutation-based scoring approaches, GRaSP, or GSTs. However, on reading some of the references I feel that this introduction is not ideal to those who aren't experts. I'm raising my score because I don't see problems, but I cannot assess the independently assess the correctness and true impact of this work without more background.

---

> > > ### Author Response · Authors · 2023-08-16
> > > **Thanks for the comments! Let me elaborate a bit...**
> > >
> > > Nice--I'm one of the other authors; the main author gave me permission to respond, haha, he will regret it... :-)
> > >
> > > Mentioning scale-freeness in the intro is a great idea. There is reasonable consensus among many now that fMRI connectivity is scale-free. GRaSP and BOSS deal with this sort of data very nicely, far better than the usual causal search algorithms, even out to an average degree of 20, without any compromise in quality. Also, we plan to rework the introduction in response to various comments.
> > >
> > > We're thinking maybe the "wow" factor of doing graphs with an average degree of 20 perhaps didn't come through well. Nearly all papers we've seen in Neurips and elsewhere deal only with very sparse graphs, average degree 2 or 4, or 6, which for a large number of variables is sparse in the extreme. We wanted to emphasize that our method could handle a much higher average degree without sacrificing accuracy (as almost all algorithms do, to the best of our knowledge) and responds well to many people's worries about doing causal searches accurately with dense graphs. DirectLiNGAM does reasonably well for dense graphs, but only under the stronger assumption of linear (strong) non-Gaussianity, and the empirical fMRI data we use are quite Gaussian, so it is not surprising (see Table 2) in our paper that DirectLiNGAM performs at chance levels for this sort of data. (Compare this to DAGMA in our charts, for instance, another Gaussian method published recently in Neurips and the best-performing continuous optimization algorithm we could find, so this is a comparison to that whole approach, so far as we currently know.)
> > >
> > > I was actually very excited to see the comment about scalability. For this paper, the state of the art for Gaussian searches for dense models is currently GRasP, though the published GRaSP scales only to about 100 variables, and in that paper, the maximum average degree tested was 10. In this paper, using a different permutation algorithm, we've increased the number of variables we can comfortably analyze 10-fold and the average degree we can analyze 2-fold without sacrificing accuracy. This sets a new state of the art for this problem, so far as we know. Notice also that even DirectLiNGAM, under a stronger assumption, takes much longer to run on this problem without appreciably better results; BOSS is able to match and even sometimes improve on the performance of DirectLiNGAM even for the linear, non-Gaussian case, which is helpful since even if you know that some variables are non-Gaussian you may not know that all variables are non-Gaussian.
> > >
> > > What excited me was that you wanted to extend the analysis to voxel-level analysis, which would to at least 40,000 variables. Genetic data also scales into that general dimension if one includes protein data in the analysis. We are keen to scale methods to that dimension for dense, scale-free data and are currently working through the various issues with parallelization, etc. However, there is no room in the current paper to address those optimizations; we are at the page limit. We plan to write another paper that does. It goes without saying that with good parallelization (not easy, key steps not embarrassingly parallel), we could jump from doing these analyses on laptops in a single thread to many-core machines.
> > >
> > > Also, we will definitely include the multiple-runs option in the methods section, thanks, and mention that for small numbers of variables (not so much for massive models), it can help. The idea of post-processing the BOSS output with a GFCI wrapper was proposed in the GRaSP paper but not explored, and we are hesitant to include any specific proposals on that before we've worked through the thorny issues of scaling a latent variable search up to the size problems that we are interested in. No one can do that now, even remotely, but we will try. We do have implementations of the procedure that people can try if they wish to see how well it does before such optimization. Using BOSS as an initial step certainly improves accuracy for these PAG methods.
> > >
> > > Regarding not being abreast of the permutation search literature, for one thing, GSTs are completely novel as of this paper, as is the BOSS algorithm, so we weren't expecting anyone to be familiar with those, no worries. The permutation methods, though, are very promising, as they currently outperform DAG (CPDAG) search over all other methods we know of for the linear Gaussian case by a long shot and match the best-performing algorithms for the linear non-Gaussian case. We're hoping to raise awareness of this methodology--i.e., hoping people will go back and read the Raskutti and Uhler paper, the Solus et al. paper, and the Lam et al. GRaSP paper and various other papers that have been published using permutations of variables orders as a key step.

---

### Author Rebuttal · Authors · 2023-08-10

We thank all reviewers for your comments! We noticed that none of the reviewers were especially confident in their reviews (none had confidence greater than 2); perhaps our responses can help in this regard.

The reviewers did not have many comments regarding the technicalities of the algorithm or data structure presented in the paper. There were, however, some areas where the presentation can be improved, and it’s clear from the comments that some context is needed to tell the reader what the contributions of the paper are to the practical issue of analyzing large-scale data accurately from a causal point of view.

We will address comments that ask for clarification or more information and add text in the paper to address these comments.

In general we plan to:

 - add a discussion of future work, limitations, and ethical considerations

 - add a discussion about complexity of the algorithm

 - add more details to the figure that demonstrates BOSS's scalability

 - clarify that GRaSP is using GSTs — without GSTs, GRaSP does not scale much past 100 variables

 - add comments about Direct LiNGAM in the Gaussian case

 - adjust language for clarity where necessary and correct any typos

---

### Decision · Program_Chairs · 2023-09-21

**Decision:**

Accept (poster)

**Comment:**

The paper extends the Greedy Relaxation of the Sparsest Permutation (GRaSP) approach. Its contribution is to design and exploit a data structure and caching method to explore the permutation space and reuse the intermediate calculations, enabling causal discovery to scale up to some 1,000 nodes for problems with very large degree (20), which is a breakthrough in the causal discovery domain.

The discussion with the reviewers showed that the paper needs a careful contextual setting about permutation-based search to appreciate the originality of the approach. Please also clarify its limitations, in particular w.r.t. the sample size.